# Genome sequencing is critical for forecasting outcomes following congenital cardiac surgery

W. Scott Watkins[1], Edgar J. Hernandez[2], Thomas A. Miller[3], Nathan R. Blue [4], Raquel Mae Zimmerman[2], Eric R. Griffiths [5], Erwin Frise [6], Daniel Bernstein [7], Marko T. Boskovski[8], Martina Brueckner [9,10], Wendy K. Chung [11], J. William Gaynor [12], Bruce D. Gelb [13], Elizabeth Goldmuntz [14], Peter J. Gruber [15], Jane W. Newburger [16], Amy E. Roberts[11,16], Sarah U. Morton [17], John E. Mayer Jr [18], Christine E. Seidman [19,20], Jonathan G. Seidman [19], Yufeng Shen[21], Michael Wagner[22], H. Joseph Yost [23], Mark Yandell [1] ✉ & Martin Tristani-Firouzi [24] ✉

While exome and whole genome sequencing have transformed medicine by elucidating the genetic underpinnings of both rare and common complex disorders, its utility to predict clinical outcomes remains understudied. Here, we use artificial intelligence (AI) technologies to explore the predictive value of whole exome sequencing in forecasting clinical outcomes following surgery for congenital heart defects (CHD). We report results for a prospective observational cohort study of 2,253 CHD patients from the Pediatric Cardiac Genomics Consortium with a broad range of complex heart defects, pre- and post-operative clinical variables and exome sequencing. Damaging genotypes in chromatin-modifying and cilia-related genes are associated with an elevated risk of adverse post-operative outcomes, including mortality, cardiac arrest and prolonged mechanical ventilation. The impact of damaging genotypes is further amplified in the context of specific CHD phenotypes, surgical complexity and extra-cardiac anomalies. The absence of a damaging genotype in chromatin-modifying and cilia-related genes is also informative, reducing the risk for some adverse postoperative outcomes. Thus, genome sequencing enriches the ability to forecast outcomes following congenital cardiac surgery.

Congenital heart defects (CHD) represent a complex class of often life-threatening disorders that affect >40,000 newborns in the U.S. annually. The prevalence of CHD is ~1 per 100 live births, with an incidence that varies according to the specific CHD lesion[1–3]. The genetic architecture of CHD has been the focus of several large-scale sequencing efforts[4–9], demonstrating that the genetic landscape of syndromic and sporadic CHD differ, with sporadic forms characterized by considerable locus and allelic heterogeneity[7]. More recently, work by the National Heart, Lung and Blood Institute (NHLBI)-funded Pediatric Cardiac Genomics Consortium (PCGC) has shown that dominantly and recessively inherited forms of CHD have distinct genetic and phenotypic landscapes, whereby dominant forms of CHD are significantly enriched for damaging variants in chromatin-modifying genes, while recessive forms are enriched for damaging variants in cilia-related biallelic genotypes and heterotaxy phenotypes[4,5,8,9].

Recent work has also demonstrated the value of genetic testing for outcomes prediction for specific types of CHD and within specific

**Table 1 | Absolute risk ratios for CHD phenotypes by gene pathway**

| Gene pathway | n | Cardiac phenotype | | | | |
|---|---|---|---|---|---|---|
| | | AVC (n = 64) | CTD (n = 934) | HTX (n = 219) | LVO (n = 647) | OTH (n = 389) |
| **Chromatin genes** (de novo) | 28 | - | 0.52 (0.32, 0.72) | - | **1.61** (1.41, 1.81) | **1.85** (1.44, 2.26) |
| **Cilia genes** (recessive) | 35 | 3.02 (0.00, 6.97) | 0.55 (0.41, 0.69) | **2.63** (2.06, 3.20) | 1.09 (0.90, 1.28) | 0.67 (0.18, 1.16) |
| **HHE genes** (de novo) | 9 | - | 0.82 (0.00, 1.74) | 1.58 (0.00, 9.05) | 0.84 (0.00, 4.88) | 1.93 (0.00, 4.29) |
| **Wnt genes** (de novo) | 18 | - | 0.54 (0.16, 0.92) | - | **2.13** (1.86, 2.40) | 0.99 (0.00, 2.25) |
| **FoxJ1 genes** (recessive) | 6 | - | 0.56 (0.00, 3.54) | **6.89** (3.30, 10.36) | 0.78 (0.00, 4.82) | 0.81 (0.00, 3.54) |
| **Notch1 genes** (de novo) | 8 | - | 1.49 (0.96, 2.02) | - | 1.36 (0.00, 2.86) | - |
| **Signal trans genes** (de novo) | 14 | 3.56 (0.00, 20.50) | 0.69 (0.24, 1.14) | - | **1.51** (1.01, 2.01) | 1.26 (0.00, 2.84) |
| **TGF-β genes** (de novo) | 13 | - | 1.11 (0.74, 1.48) | - | 1.09 (0.32, 1.86) | 1.37 (0.00, 3.05) |
| **CHD genes** (de novo) | 51 | - | 0.86 (0.78, 0.94) | 0.27 (0.00, 1.43) | **1.22** (1.11, 1.33) | **1.59** (1.38, 1.80) |

Each ratio is reported as the mean and 95% confidence interval from 1,000 bootstrap replicates fitted to a t-distribution. Absolute risk ratios with 95% CIs > 1.00 are bolded. The column labeled *n* indicates the number of patients with damaging genetic variants/genotypes (GEM score ≥1) found in that gene pathway. The total number of patients in each CHD category is listed in the column heading. Dashes indicate no patients with damaging genotypes. Phenotype categories are atrioventricular canal defects (AVC), conotruncal defects (CTD), heterotaxy/laterality defects (HTX), left ventricular outflow tract obstructions (LVO), and all other defects (OTH). Additional abbreviations: HHE, high heart expression genes in the developing mouse heart; CHD genes, a curated list of genes previously reported to cause CHD (see also Supplementary Data 3).

clinical contexts[10–14]. Broader investigations, however, have faced difficulties in assaying genetic contributions across multiple CHD phenotypes and clinical contexts, in part due to the widely varying severity of CHD lesions and the complex medical and surgical interventions necessary for survival.

Here, we demonstrate that condensing heterogenous CHD phenotypes into clinically relevant phenotypic categories using anatomic descriptors[15] renders these data amenable for outcomes analyses. We also show that the high allelic and locus heterogeneity characteristic of CHD can be overcome using an artificial intelligence (AI) genome interpretation tool[16], followed by categorization of damaging genotypes into molecular pathways or gene categories. This two-pronged approach of phenotypic and genotypic classification, when combined with probabilistic graphical models, enables clinically relevant and highly personalized risk estimates in patients undergoing congenital cardiac surgery.

## Results

### Refining the genetic architecture of CHD

For genetic and outcomes analysis, the prospective observational cohort study population consisted of 2,253 PCGC probands (1,998 trios, 21 duos, 234 singletons) with exome sequencing, CHD phenotypes, and surgical outcomes data. The AI-based genome analysis tool GEM[16] identified putative damaging genotypes in 238 participants (10.6% of the cohort). A total of 131 damaging de novo/dominant and 196 damaging recessive/biallelic genetic variants were discovered (Supplementary Data 1, 2). There were 17 genes with damaging de novo variants in two or more patients. The most commonly recurrent de novo variants were in known CHD-related genes such as *KMT2D* (11), *CHD7* (6), *RAF1* (3), *JAG1* (3), and *TAB2* (3). Biallelic damaging genotypes were observed in multiple patients for several genes including *DYNC2H1* (3), *DNAH5* (3), *LAMA2* (3), *GDF1* (2), and *IFT140* (2).

We discovered that CHD phenotype categories were enriched for damaging genetic variants in specific gene pathways/categories (Table 1, Supplementary Data 3). For example, the LVO class was enriched 1.6-fold (CI 1.4–1.8) for damaging de novo genotypes in chromatin-modifying genes, with this signal driven primarily by patients with hypoplastic left heart syndrome (HLHS), a subset of LVO in which these genotypes showed a 1.9-fold (CI 1.3–2.5) enrichment. While previous studies implicated damaging chromatin-modifying gene variants in CHD cohorts at large[4,5,8,9], our analyses here help to define the specific CHD subtypes most influenced by damaging variants in chromatin-modification genes. The LVO phenotype class was also enriched for de novo genotypes in *WNT* genes (2.1-fold, CI 1.9–2.4), signal transduction genes (1.5-fold, CI 1.0–2.0), and a curated

list of genes known to cause CHD (1.2-fold, CI 1.1–1.3; see Supplementary Data 1, 2, and 3). Notably, damaging genotypes in these pathways were not enriched in HLHS patients, although an association might be detectable with a larger sample size.

The HTX phenotype class was enriched for damaging recessive/biallelic variants in cilia-related genes (2.6-fold, CI 2.0-3.2) and showed proportionally higher enrichment in the subset of motile cilia genes modulated by *FOXJ1* (6.9-fold, 3.3-10.4), findings consistent with previous reports[5,8,17] Patients with *FOXJ1* pathway mutations accounted for four of nine HTX patients (44%) in the cilia enrichment subset. Damaged genes in the *FOXJ1* pathway were *ARMC4*, *CCDC151*, *DNAI1*, *DRC1*, *IFT172*, and *SPEF2*. The heterogenous OTHER (OTH) phenotype class was enriched for damaging de novo variants in chromatin-modifying genes (1.9-fold, CI 1.4-2.3) and the curated CHD genes (1.6-fold, CI 1.4–1.8). While the chromatin and CHD gene lists share 35 genes, the chromatin-modifying genes did not account for all of the association signal. For example, damaging genotypes in the curated CHD list associated with the OTH category included Noonan syndrome genes SOS1 (2), RAF1 (2), and BRAF. Consistent with previous findings, the AVC category showed an association with cilia genes but did not reach statistical significance, likely due to a low number of patients with AVC. The patients with CTD did not show a significant association with these gene lists, but our data set did not include copy number variants (CNVs), which are known to show association with CTD defects and tetralogy of Fallot[18].

### Damaging genotypes impact surgical outcomes

To further explore the relationships between genetic and clinical variables, we utilized Bayesian networks, a powerful statistical framework that can model complex dependencies, including non-linear relationships and indirect associations, in a probabilistic manner. In Bayesian networks, variables are depicted as nodes in a graph and conditional dependencies between variables are represented by the edges connecting those nodes. Once a network is constructed, the impact of any combination variables on any selected outcome can be quantified, while controlling for the effects of other variables incorporated into the network. The Bayesian networks describing the conditional dependencies among damaging genotypes, CHD phenotypes and post-operative variables are shown in Fig. 1a, b.

Damaging genotypes in chromatin-modifying and cilia-related genes (defined by a GEM[16] score ≥1.0) increased the probability of severe adverse clinical outcomes following congenital cardiac surgery, including mortality, cardiac arrest, and prolonged mechanical ventilation (>7 days post-surgery). For example, damaging de novo chromatin genetic variants increased the probability (relative risk ratio) of

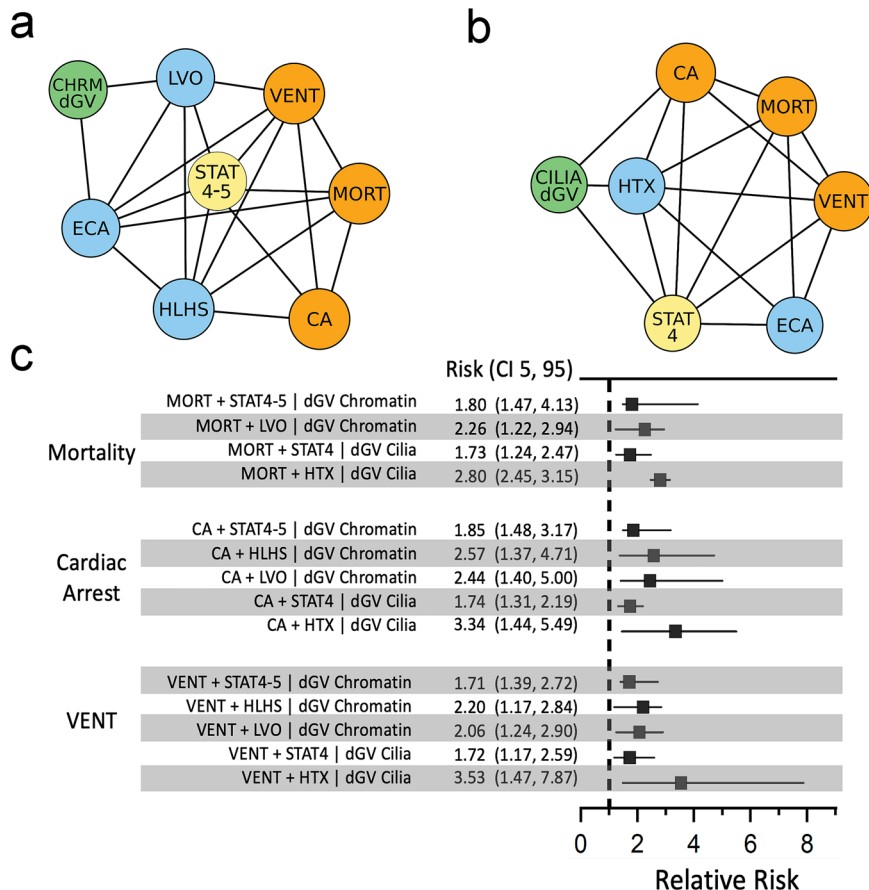

**Fig. 1 | Damaging chromatin and cilia genotypes predict adverse post-operative outcomes in the context of CHD phenotypes and surgical complexity.** Bayesian networks display a best machine-learned relationship among genotypes, phenotypes, and outcomes for 2253 surgical patients with CHD. Each network node represents a present/absent variable. Damaging genotypes in chromatin-modifying genes (CHRMdGV) or cilia-related genes (CILIAdGV) were identified from the exomes of 2253 CHD patients by GEM. Phenotype classes were predicted from Fyler codes using XGBoost. Surgical outcomes for each patient were obtained from the Society of Thoracic Surgeons national database. Relative risks for selected adverse surgical outcomes were then estimated from each network using network-propagated probabilities. **a** An exact Bayesian network depicting the relationship among damaging de novo genetic variants in chromatin-modifying genes (green), phenotypes: LVO, HLHS, and ECAs (blue), surgical STAT4 or STAT5 category (yellow), and adverse surgical outcomes (orange). **b** An exact Bayesian network depicting the relationship among damaging recessive genetic variants in cilia-related genes (green), phenotypes: laterality defects (HTX) and extra cardiac anomalies (ECAs) (blue), surgical STAT4 category (yellow), and adverse surgical

outcomes: long ventilation time, cardiac arrest, and mortality (orange). Directed acyclic graphs were moralized and displayed as non-directional networks. **c** Relative risk ratio estimates for adverse post-operative outcomes and CHD phenotypes or surgical complexity, comparing probands with and without damaging genotypes. Empirical ninety-five percent confidence intervals (CI 5, 95) are based on 1000 resampled network-based probability estimates. Because the resampling distribution estimates may be constrained by the Bayesian network structure, error bars may not be symmetric with respect to the median point estimate (see Methods). Abbreviations: CHRMdGV - de novo damaging genotypes in chromatin-modifying genes, LVO - left ventricular outflow tract obstruction, HLHS - hypoplastic left heart syndrome, CILIAdGV - biallelic damaging genotypes in cilia-related genes, ECA - extra cardiac anomaly, HTX - heterotaxy/laterality defects, MORT - mortality, STAT4 - surgical STAT4 category, STAT4-5 - surgical STAT 4 or STAT5 category, VENT - post-operative ventilation time >7 days. For Figs. 1c, 2, and 3, there were 8–35 patients in the conditional subsets and 1–5 patients in the target sets (see Supplementary Data 4).

mortality 1.8-fold (CI 1.5-3.2), cardiac arrest 1.7-fold (CI 1.4-2.9) and prolonged ventilation 1.6-fold (CI 1.4-2.3). Likewise, damaging recessive/biallelic cilia genotypes increased the probability of mortality 1.4-fold (CI 1.1–2.1), cardiac arrest 1.5-fold (CI 1.1–2.3) and prolonged ventilation 1.4-fold (CI 1.1–2.0). Reciprocally, in the context of these networks, the absence of a damaging genotype was associated with lower risk for these adverse post-operative outcomes, as compared to patients with a damaging genetic variant. Thus, for a proband without a damaging de novo chromatin genotype, the relative risk ratio for mortality was 0.55 (CI 0.31-0.69), 0.55 (CI 0.34-0.73) for cardiac arrest and 0.61 (CI 0.44-0.72) for prolonged ventilation. For a proband without a damaging recessive/biallelic cilia genotype, the relative risk ratio for mortality was 0.72 (CI 0.48-0.91), 0.63 (CI 0.43-0.88) for cardiac arrest and 0.70 (CI 0.51–0.92) for prolonged ventilation. The patient counts for the relative risk ratios presented here range from 1

to 179 (Supplementary Data 4). While the number of adverse events in some of these genetic and clinical contexts was relatively low, the Bayesian statistical framework is particularly well-suited for predictions when numbers are limiting[19,20], which was our motivation for employing this framework.

## Damaging genotypes impact surgical outcomes in the context of surgical mortality risk category

We discovered that damaging chromatin and cilia genotypes were associated with an increased risk of mortality for probands undergoing the highest risk surgical procedures. Thus, probands who died after a STAT4 or STAT5 surgical procedure were 1.8-fold (CI 1.5–4.1) more likely to harbor a damaging chromatin variant. Similarly, those who died after a STAT4 surgery were 1.7-fold (CI 1.2–2.5) more likely to harbor a damaging recessive/biallelic cilia genotype. Damaging

chromatin and cilia genotypes were overrepresented in probands experiencing cardiac arrest or prolonged mechanical ventilation following the most complex surgical procedures (Fig. 1c).

## Damaging genotypes impact surgical outcomes in the context of CHD phenotypes

More broadly, considering mortality in the context of CHD phenotypes, LVO patients who died were 2.3-fold (CI 1.2–2.9) more likely to harbor a damaging de novo chromatin genotype, while HTX patients who died were 2.8-fold (CI 2.5-3.2) more likely to harbor a damaging recessive/biallelic cilia genotype. Similarly, damaging chromatin or cilia genotypes were overrepresented in probands with LVO, HLHS, and HTX who experienced cardiac arrest or prolonged post-operative ventilation (see Fig. 1c). Specifically, HTX patients who arrested post-operatively were 3.3-fold (CI 1.4-5.5) more likely to harbor a damaging recessive/biallelic cilia genotype, compared to similar patients without a damaging cilia genotype. Collectively, these findings demonstrate that genome sequencing data are critical for predicting severe postoperative events in the context of specific CHD phenotypes and the highest risk congenital heart surgeries.

## Damaging genotypes impact surgical outcomes in the context of extracardiac phenotypes

Given the recognized impact of extra cardiac anomalies (ECAs) on outcomes following congenital cardiac surgery[10,14,21], we also explored the relationship between ECAs and adverse post-operative outcomes in the context of genotypes and CHD phenotypes for 898 patients with reported ECAs (Supplementary Data 5 and 6). ECAs increased the probability (relative risk) of mortality 2.8-fold (CI 1.5–2.9) and prolonged ventilation 1.7-fold (CI 1.6–1.7) following congenital cardiac surgery. Consistent with previous findings[10], all damaging de novo genetic variants and de novo variants in chromatin-modifying genes were enriched in probands with ECAs 1.47-fold (CI 1.44-1.48) and 2.09-fold (CI 2.08-2.11), respectively. By contrast, damaging recessive/biallelic cilia genotypes were not enriched in probands with ECAs (0.8-fold; CI 0.8–1.1).

We also examined reciprocal effects, i.e., the impact of predicted damaging genotypes on ECAs and adverse outcomes. For example, probands with damaging de novo chromatin genotypes identified by GEM were 2.5-fold (CI 2.2–5.0) more likely to have an ECA and die, compared to probands without a damaging chromatin genotype, and 2.4-fold (CI 2.0-3.4) more likely to have an ECA and prolonged ventilation (Fig. 2). Moreover, a damaging recessive/biallelic cilia genotype identified by GEM increased the probability of mortality in probands with an ECA 1.5-fold (CI 1.0–2.8), compared to similar probands without a damaging cilia genotype, and increased the probability of prolonged ventilation in the presence of an ECA 1.5-fold (CI 1.1–2.5). Additionally, a damaging cilia genotype increased the probability of prolonged ventilation in HTX patients with an ECA 4.0-fold (CI 1.7-10.6), compared to similar patients without a damaging cilia genotype (see Fig. 2). Taken together, these findings demonstrate that damaging genotypes in chromatin and cilia genes increase the likelihood of severe post-operative events in the setting of ECAs.

The number of probands experiencing adverse outcomes in the AVC, CTD, and OTHER categories and harboring damaging gene pathway variants was too low to warrant generation of Bayesian networks for outcomes prediction in these CHD phenotypes. The reasons for this are multi-factorial, including genetic and phenotypic heterogeneity, low number of patients in some categories, as well as excellent surgical outcomes in these categories. For example, no patient with AVC died post-operatively in this cohort. Consequently, larger cohorts with complete genome information, including copy number and structural variation changes, are necessary to adequately predict the impact of genetics on outcomes for these CHD phenotypes. However, damaging genotypes in several gene pathways/categories, such as

FOXJ1-controlled genes, high murine heart expression genes (HHE), WNT signaling genes, NOTCH signaling genes, and genes in a curated CHD gene list were predictive of mortality for the most complex surgical categories (Fig. 3). Damaging genotypes in signal transduction and TGF-β pathways were not predictive of mortality in this data set. Taken together, these findings highlight the value of genomic data for predicting adverse outcomes following congenital cardiac surgery, especially in the context of CHD phenotypes, ECAs and surgical complexity.

## Discussion

Assessing the impact of genetics on patient outcomes in CHD is complicated by the intrinsic severity of the cardiac lesion, the complex medical and surgical interventions necessary for survival, and the high degree of phenotypic, locus and allelic heterogeneity. The NHBLI-funded PCGC is one of the world's largest collections of genetic, phenotypic, and clinical variables for CHD and thus provides an excellent resource for exploring the utility of genomics data for outcomes prediction. In this study, we implemented an explainable AI-based analysis framework to automatically classify CHD patients into phenotype categories and identify damaging genetic variants and genotypes. This approach allowed us to explore how damaging genotypes impact outcomes following congenital cardiac surgery, in the context of specific CHD phenotypes, ECAs, and surgical complexity, providing risk estimates for specific clinical contexts.

Overall, the number of adverse events in probands with damaging genotypes in some clinical contexts was relatively low, despite an initial corpus of >2000 cases. It is well established that the Bayesian statistical framework is particularly well-suited for predictions when numbers are limiting[22]; indeed, this was our motivation for employing this analytical framework. Bayesian approaches allow for the incorporation of prior knowledge, explicitly model uncertainty, and provide established best-practice methodologies to avoid overfitting. Moreover, Bayesian methods model uncertainty using a probability distribution (the posterior distribution) for each parameter rather than a single point estimate. This feature is particularly valuable in low-data scenarios, allowing the model to express uncertainty through conditional probability distributions even with limited data[23]. The confidence intervals we report were generated by simultaneously resampling the data and rebuilding the net. This is a gold standard approach for estimating uncertainty when data are limiting. In general, the wide confidence intervals reflect this data scarcity. Taken together, these features allow Bayesian networks to make better informed predictions in low-data scenarios than do traditional frequentist approaches[22–24]. It should be noted that risk projections for some of the rare, high-risk scenarios are rough estimates. While future analyses using larger cohorts will one day refine these estimates, our results provide a necessary starting point for that future work.

De novo variants associated with CHD are enriched in genes related to chromatin regulation[4,5,8,9]. Our results identify LVO lesions and confirm HLHS as a principal driver of the chromatin signal in this cohort. HLHS is one of the most severe forms of CHD and associated with substantial morbidity and mortality. Our results show that the subset of HLHS patients with damaging genetic variants in chromatin genes has even greater risk (up to 2.6-fold) for severe post-operative outcomes in the context of the most complex surgical procedures.

Our findings also reinforce previous studies showing that damaging recessive/biallelic genotypes in cilia-related genes are overrepresented in the HTX/laterality phenotype category[5,8]. Notably, genes identified in the *FOXJ1* pathway were proportionally higher in HTX patients where four of six gene findings were supported by recessive mouse models[25] of CHD with heterotaxy (*ARMC4, CCDC151, DNAI1, DRC1*). All six of the human genes are linked to cilia dysfunction and human primary ciliary dyskinesia. Our results here demonstrate the additional utility of genetic findings for outcomes predictions.

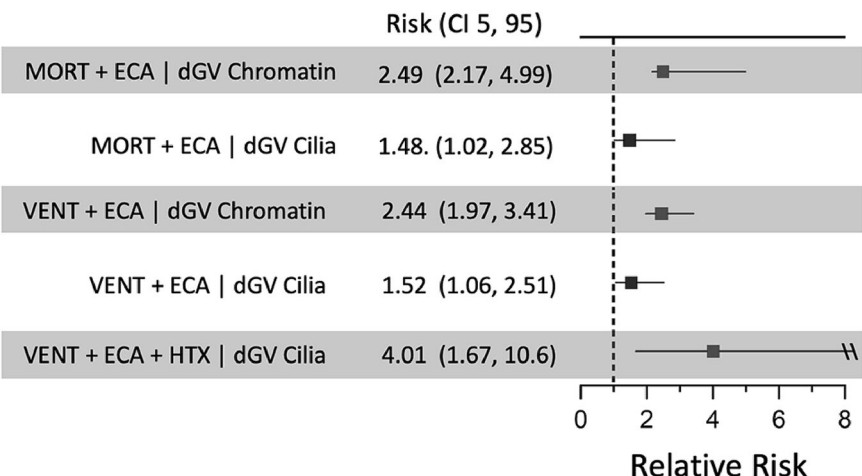

**Fig. 2 | Damaging chromatin and cilia genotypes predict adverse post-operative outcomes in the context of extracardiac anomalies.** Relative risk ratios for adverse post-operative outcomes and extracardiac anomalies (ECAs), comparing probands with and without damaging genotypes in chromatin-modifying or cilia-related genes. Each risk estimate shows the point estimate of the network propagated relative risk. Empirical ninety-five percent confidence intervals (CI 5, 95) were generated by resampling the data matrix with replacement and re-estimating the network propagated risk 1000 times. Because the resampling distribution estimates may be constrained by the Bayesian network structure, error bars may not be symmetric with respect to the median point estimate. Target and conditional counts are listed in Supplementary Data 4. The \\ symbol represents a (CI 5, 95) that exceeds the x-axis range.

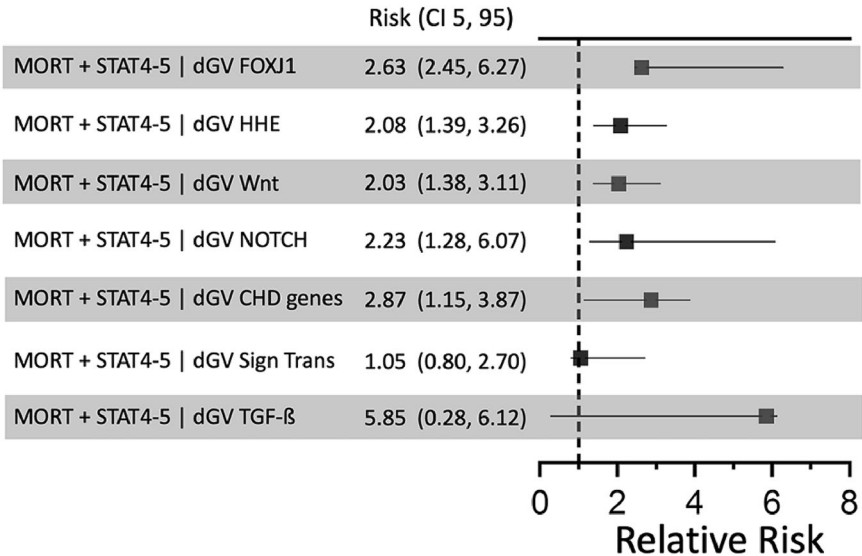

**Fig. 3 | Damaging genotypes in various gene categories/pathways are predictive of mortality for the most complex surgical procedures.** Relative risk ratios for adverse post-operative outcomes and surgical complexity compare probands with and without damaging genotypes in various gene pathways or categories. Gene lists are described in Supplementary Data 3 and have been previously published[5,8,9]. There is overlap between gene lists, with some genes represented in more than one gene pathway/category (Supplementary Fig. 4). Each estimate shows the point estimate of the network propagated relative risk. Empirical ninety-five percent confidence intervals (CI 5, 95) were generated by resampling the data matrix with replacement and re-estimating the network propagated risk 1,000 times. Because the resampling distribution estimates may be constrained by the Bayesian network structure, error bars may not be symmetric with respect to the median point estimate. Target and conditional counts are listed in Supplementary Data 4.

Damaging recessive/biallelic cilia genotypes increase the risk of severe adverse post-operative outcomes in the context of surgical complexity, HTX phenotype and the presence of an ECA. For example, damaging recessive/biallelic cilia genotypes substantially increase (4-fold) the risk of prolonged ventilation for HTX patients with an ECA. These findings are consistent with an emerging body of literature implicating cilia dysfunction, HTX, and respiratory complications following congenital cardiac surgery[26,27].

Established and emerging literature has highlighted the impact of genetics on mortality and other adverse outcomes following congenital cardiac surgery, mostly focusing on the impact of copy number variants[10–14]. Damaging de novo genic variants were associated with worse transplant-free survival and longer times to final extubation in a previously reported subset of the PCGC cohort ($N = 1268$)[10]. Here, we expand upon these findings in the largest study to date relating genotypes to CHD surgical outcomes. Our analyses reveal that damaging genotypes in specific gene pathways/categories impact post-operative outcomes across CHD phenotypic categories in specific and quantifiable ways.

Our AI approach allowed us to unravel the conditional dependencies among diverse clinical and genetic variables and to discover their impacts, either in isolation or in combination, on post-operative

outcomes. These findings define a critical role for genome sequencing in outcomes prediction for congenital cardiac surgeries, especially in the context of higher risk surgical procedures, specific CHD phenotypes and ECAs. Importantly, the absence of damaging genotypes was associated with lower predicted probability of adverse outcomes following congenital cardiac surgery as compared to those patients with damaging genotypes. Thus, genomic information is informative whether or not a proband has an identified damaging genotype.

The ability to quantify the risk of adverse outcomes allows for crucial, early deployment of potential therapeutic strategies to mitigate these risks and improve patient outcomes. For example, an extensive body of literature links ciliary dysfunction with respiratory complications, including prolonged ventilation, in the post-operative period[27–32] Pre-operative knowledge of a damaging cilia genotype would allow for early institution of aggressive airway clearance therapy, mucolytic therapies, inhaled β2 agonists and specific ventilation strategies to promote mucociliary clearance[31]. Avoiding common anesthetic and analgesic agents that are known to impair mucociliary function may also be beneficial in these patients[32]. As turn-around times shorten and costs decline for ultra-rapid and rapid WGS, genomic data will be available in the pre-operative period to allow for risk stratification and early interventions to mitigate adverse post-operative outcomes.

Nevertheless, there are limitations inherent to this study. For example, the PCGC population is not an inception cohort and thus is likely depleted for genetic lesions that predispose to early death, meaning our morbidity estimates are likely lower bounds. Although the PCGC cohort reflects a broad spectrum of CHD, recruitment of severe CHD forms was favored, leaving us under-powered to investigate the impact of genome sequencing for less severe CHD phenotypes. Additionally, while large clinical registries, such as the STS database, are invaluable resources for outcomes research, these databases, despite the inclusion of auditing features, may suffer from data quality issues, variability in the abstraction of data, batch effects, and missing data[33–35] that might impact the interpretation of the results presented here. Finally, we acknowledge that replication in an independent, external cohort is important for reproducibility and scientific rigor. However, replication is not possible or directly comparable, as no other CHD cohorts in the world have the same depth or breadth of post-surgical variables with genomic information. The lack of comparable cohorts reflects the current constraints of the field and underscores the unique nature of the PCGC dataset. While replication in an independent cohort is not currently feasible, the relationships captured by the Bayesian network model are biologically and clinically plausible, and importantly, align with existing literature. For example, the risk of prolonged mechanical ventilation in patients with damaging cilia genotypes in the context of HTX and STAT4 surgeries is well-supported by literature linking cilia dysfunction and HTX with post-operative respiratory complications[27–30]. As more data become available in external CHD cohorts, we plan to validate our model prospectively.

Looking to the future, a more complete description of the genetic and outcomes landscape of CHD could be enabled through clinical genome sequencing of CHD patients at even greater scales, together with initiatives by major consortia to collect and distribute genomic and clinical data more broadly. As whole genome sequencing supplants exome sequencing, and as new technologies, including long-read sequencing, reference-free whole genome assembly, and sequence analysis of somatic tissues are more broadly implemented, the diagnostic yield for CHD is likely to improve. Given the rapid decline in costs, the increasing availability and quick turn-a-round time, rapid genome sequencing is now poised to become the standard of care for all critically ill newborns[36,37]. Our findings make it clear that genome sequencing of all newborns with complex CHD will empower personalized risk-stratification for outcomes following congenital cardiac surgery.

## Methods

### Study participants

All patients in the prospective observational cohort study were diagnosed, phenotyped, and recruited by PCGC centers and participating regional hospitals into the PCGC Congenital Heart Disease Network Study (CHD GENES: ClinicalTrials.gov identifier NCT01196182; https://clinicaltrials.gov/). Written informed consent was obtained from all participants or the participants' guardians. Approval for this research was obtained by the institutional review boards of participating centers, including Boston's Children's Hospital, Brigham and Women's Hospital, Great Ormond Street Hospital, Children's Hospital of Los Angeles, Children's Hospital of Philadelphia, Columbia University Medical Center, Icahn School of Medicine at Mount Sinai, Rochester School of Medicine and Dentistry, Steven and Alexandra Cohen Children's Medical Center of New York, Lucile Packard Children's Hospital Stanford, University of California-San Francisco, University of Utah, and Yale School of Medicine. Automated CHD phenotype classification was performed on 14,765 PCGC participants. A subset of these participants who had both exome sequencing and perioperative data (2,253 total patients; 1323 males, 930 females) was used for network analyses.

### Clinical phenotypes

Cardiac diagnoses were obtained from review of echocardiogram, cardiac MRI, catheterization, and operative reports at the time of enrollment into the PCGC[4,5]. Detailed cardiac diagnoses for each patient were coded using the Fyler system[15]. Extra-cardiac anomalies (ECAs) were identified at the time of PCGC enrollment[4,5] (see Supplementary Data 5 and 6). Any structural anomaly that was not acquired was classified as an extra cardiac anomaly (ECA).

### Post-operative variables

For patients undergoing open heart surgery, surgical and hospitalization data were obtained from PCGC participating centers using the local data collected for submission to the Society of Thoracic Surgeons Congenital Heart Surgery Database (STS-CHSD)[38]. All surgeries were standard of care. A total of 59 surgical complication variables were extracted for analysis. The size of the final data set was constrained to 2253 patients, such that all patients had WES and surgical variables had no more than 10% missing data. Most patients had multiple cardiac surgeries. A patient was scored as having an adverse event or surgical complication (e.g., prolonged mechanical ventilation) if that event occurred for any surgery at any age.

### Surgical complexity

Surgical complexity is a well-known driver of mortality and morbidity. In response, the STS-European Association for Cardio-Thoracic Surgery (STAT) has created risk assessment categories in which procedures are grouped based on similar mortality rates[39]. STAT categories range from 1 to 5, with STAT1 representing the procedures with the lowest mortality rates and STAT5 representing the procedures with the highest mortality rates.

### CHD classification

The PCGC has classified cardiac diagnoses for over 14,000 CHD probands using the Fyler coding system, which describes the congenitally malformed heart using a vocabulary of >3000 possible phenotypic descriptors[15]. While this system allows for highly granular descriptions of heart defects, we hypothesized that condensing these terms into a few clinically relevant phenotypic categories might render them more tractable for outcomes analyses. Thus, we sought to automate cardiac phenotype classification across the entire PCGC cohort, assigning each patient to a single category. To do so, we used five major cardiac categories derived from a previous PCGC study[5]: left ventricular outflow tract obstructions (LVO), laterality and heterotaxy defects (HTX), atrioventricular canal defects (AVC), conotruncal defects (CTD), and

other defects (OTH), which includes simple atrial septal defects and more complex heart defects not assigned to the other four categories[5,15]. Each participant was assigned uniquely to one of the five phenotypic categories.

Considerable heterogeneity exists among patients' heart phenotypes and the five phenotype categories. For example, pulmonary stenosis and ventricular septal defects occur in all five phenotype categories. The observed rate of each Fyler code for each phenotype category in the 2,253 CHD analysis individuals is shown in Supplementary Data 7. This heterogeneous data structure suggested that a supervised learning model that leverages prior physician classification information[5] and weights each cardiac defect based on expert knowledge would perform better than an unsupervised classification approach. Therefore, we elected to use a gradient-boosted decision tree model trained on physician-based classification of 3,000 patients to learn the importance (i.e., information gain) for each Fyler code. Once trained, the algorithm can probabilistically assign each proband to one of the five phenotype categories. In the final algorithm, Fyler codes strongly associated with a specific phenotype category in the training data greatly increase the probability that the patient belongs to a specific category. For example, tetralogy of Fallot is nearly always assigned to CTD, while hypoplastic left heart is highly predictive of LVO. In contrast, pulmonary stenosis has lower predictive value for classification.

Model learning and classification was performed with an ensemble-based method using the julia XGBoost library (v1.5) [https://github.com/dmlc/XGBoost.jl] for XGBoost [https://xgboost.ai][40]. The truth set for training the classifier included 3,000 CHD patients, 2,752 PCGC patients previously assigned into the five CHD categories[5] and 248 randomly selected PCGC patients that were manually reviewed and assigned to a CHD phenotype category. A gradient-boosted probabilistic patient classifier was built with XGBoost[40] using Fyler terms from the 3,000 CHD patients (Supplementary Data 8). Model training was performed with five-fold cross-validation (CV) and replacement subsampling using a grid search method to minimize mlogloss, std-mlogloss, and classification error (Supplementary Fig. 1a). Information gain and frequency for the top 20 Fyler codes in the model is shown in Supplementary Fig. 1b. The model provides an estimate of the probability of inclusion to each of the five phenotype classes for each patient, and the highest probability is used to assign each patient to a final phenotype category (Supplementary Fig. 2). Prediction confidence was assessed by comparing the most probable class assignment to the second-best assignment (Supplementary Fig. 3a, b). Accuracy was assessed by comparing the patient's known phenotype category, derived from expert knowledge[5], to the patient's predicted phenotype category label (Supplementary Data 9). Single-class prediction accuracy for the training data was higher for HTX (98.9%), AVC (97.8%), and LVO (97.7%) than for CTD (95.3%) and OTH (91.9%), where a low level of ambiguity occurred. Overall CV training classification accuracy was 97.7% with a specificity of 99.3%, and sensitivity of 97.7% (Supplementary Data 10). We then applied the trained classifier to 14,765 PCGC CHD patients with Fyler descriptors to assign each patient to one of the five phenotype categories (Supplementary Data 11). Final classification of the 3,000 training patients was identical to the original training predictions. The five phenotype categories remained generally proportional between the training data and the full data set, with a maximum observed difference of 4.2% for CTD patients.

**AI-based scoring of predicted damaging genetic variants**
All CHD patients were sequenced using exome capture (Agilent, Illumina) and sequenced on Illumina sequencing platforms using 100 bp paired-end reads. Read data was aligned using the human reference genome (GRCh37) and then genotyped with GATK / Sentieon using best practice recommendations to produce a joint-called VCF file with all probands and their parents, if available. AI-based identification of candidate disease-causing genotypes was performed using Fabric GEM[16,41–44] (Fabric Genomics, Oakland, CA). GEM evaluates all genotypes present in a VCF file for a patient. GEM incorporates Human Phenotype Ontology (HPO) terms, sex, genotype frequency (gnomAD), evolutionary conservation, Online Mendelian Inheritance in Man (OMIM), gnomAD, and ClinVar information and parental genotypes in a probabilistic AI framework to identify the most likely genetic variant or genotype that explains the patient's disease phenotype. HPO terms utilized in the GEM analyses were based on each patient's Fyler phenotypes, which were mapped to HPO terms using the Clinithink software package (Clinithink, London). Notably, 48 of the 131 de novo variants identified by GEM were documented as pathogenic or likely pathogenic in ClinVar, and 27 of 48 (56%) of these were missense variants. The remaining 83 de novo variants were not listed in ClinVar, with 33 variants resulting in frameshifts, altered splice sites, or caused stop-gains, and 47 of 83 (57%) resulting in missense variants (see Supplementary Data 1). Because GEM is phenotype aware, reported genes have some support in the literature for association with the proband's particular CHD phenotype. However, we cannot exclude the possibility that in some cases, the reported damaging genotype is incidental to CHD etiology; however, the high score threshold used for these analyses (≥1), means that there was strong phenotypic support for the genotype, and/or the variant(s) involved have strong ClinVar associations with pathogenicity. Because WES are difficult substrates for copy number variant (CNV) calling, we restricted our analyses to SNVs and short indels.

GEM's gene scores are $\log_{10}$ transformed Bayes factors[45] that summarize the relative support for the hypothesis that the prioritized genotype damages the gene in which it resides and explains the patient's phenotype versus the hypothesis that the variant neither damages the gene nor explains the patient's phenotype. We used a stringent GEM score of ≥1.0 to represent a likely pathogenic genotype. A recent genomic analysis of critically ill newborns showed that a GEM score of ≥1.0 identified 90% of all true positive damaging variants, with a median of two candidate variants per patient[16]. Gene penetrance for GEM calculations was set to 0.95 to enforce strict consideration of known dominant and recessive disorders. For downstream analyses, damaging genetic variants were classified as de novo/dominant, or recessive/biallelic variants based on their inheritance pattern in trios. Dominant and de novo damaging variants were required to have a frequency of <1/10,000 in gnomAD databases (v2.1, v3.1) and most of these variants were not observed in gnomAD. Overall, we identified damaging de novo or recessive genotypes in 10.56% of the study cohort (see Supplementary Data 1, 2), in line with previous studies that utilized different methods of defining pathogenicity[4,5,8,9]. Damaging genetic variants were assigned to several functional gene pathways. The CHD gene list represents genes from the literature known to directly cause or be associated with CHD[8]. The CHD gene list contains 35 genes in the chromatin list, but no genes from the cilia gene lists and represents a diverse set of genes not specific to a single pathway. Gene lists for all other gene pathways were obtained using the reactome pathway browser (see Supplementary Data 3) and have been previously described[5,8,9] There is some overlap between gene lists, with some genes represented in more than one gene pathway/category (Supplementary Fig. 4).

Probabilistic graphical models. Probabilistic graphical models (PGMs) provide a robust explainable AI methodology capable of discovering and quantifying additive and synergistic effects amongst broad classes of variables. For the work presented here, we used a form of PGMs known as a Bayesian networks[46]. Bayesian networks are fully transparent, and their graphical representation offers an intuitive, visual, and qualitative mechanism for understanding the probabilistic dependencies among variables and the impacts of multiple variables on outcomes of interest[47–54]. Moreover, Bayesian networks offer practical advantages over regression approaches, by capturing the

entire joint probability distribution of the data, quantitatively encompassing all interrelationships among the variables incorporated in the non-linear model. Thus, a single network can be used to explore any combination of variables as a target outcome in one query and then as a risk factor for a different target outcome, all within the same model. For more on these points see[46,55].

## Feature selection

Single variables, such as damaging genetic variants in chromatin-modifying genes, were tested for conditional dependency with phenotype variables using exact Bayesian networks. Each conditional probability (e.g., the probability of LVO given a damaging de novo chromatin variant: P (LVO | *chromatin dGV*)) was estimated as the median conditional probability from 1000 independent networks. Conditional probability estimates were divided by the baseline probability for each respective phenotype to obtain absolute risk ratios. Associations with absolute risk ratios $\geq 1.0$ were selected for further analyses. Surgery-related variables associated with gene categories and CHD phenotypes were identified in a similar way. Each surgical feature was tested individually as a conditional variable with genetic variables and each of the five specific CHD phenotypes.

## Network construction

Bayesian networks were created for each CHD phenotype category. Each network included genetic and surgical variables identified in the feature selection stage above. Genetic, phenotypic, and surgical variables included or excluded for the networks and their baseline frequencies are listed in Supplementary Data 12. Final network variables are shown in Supplementary Data 13. All-cause mortality was included in each network. All input conditional variables were coded as presence/absence. A small number of missing surgical values (<10% for any single variable) was imputed using a K-nearest neighbors approach ($k = 10$). Because highly correlated variables may influence Bayesian network structure learning and risk estimation, variables included in the networks were screened for colinear and multicollinear states by correlation analysis (Supplementary Fig. 5a, b). Final network variables were not colinear and pairwise correlations were between -0.11 and 0.30, except for LVO and HLHS. The structure of each network was learned with the Silander-Myllymaki exact algorithm with Bayesian information criterion (BIC) scoring[56]. Posterior probabilities were network propagated using exact inference. The accuracy of network inferred risk estimates can be affected by sample size. Many studies advocate Bayesian analysis for low sample size applications[19,20,22–24,57]. With low sample sizes, accurate posterior probabilities and risk estimates are highly dependent on unbiased prior probability estimates. Our initial (prior) probability estimates for variables used in the study are based on 2253 CHD patients from multiple surgical sites and are, therefore, likely to be representative probability estimates for a critical CHD surgical cohort and without strong biases. Network structure learning and belief propagation were performed in R with the bnstruct and gRain R packages[58,59] and implemented in the BayesNetExplorer package [https://github.com/ScottWatkins/BayesNetExplorer].

## Risk calculations

We used two risk ratios (RR) to summarize our risk estimates, absolute RR and relative RR. For example, the absolute RR of a LVO phenotype given a damaging genotype in a chromatin-modifying gene is as follows: $Absolute\ RR_{LVO|dGVchromatin} = \frac{P(LVO=true|dGV\ chromatin=true)}{P(LVO=true)}$ which estimates the probability of LVO given a damaging mutation in chromatin-modifying genes compared to the marginal probability of LVO in the entire population. The relative RR estimates the relative change in mortality risk for LVO patients with damaging mutations in chromatin-modifying genes, compared to similar patients without a damaging chromatin genotype: $Relative\ RR_{mortality+LVO|dGV\ chromatin} = \frac{P(mortality=true,LVO=true|dGV\ chromatin=true)}{P(mortality=true,LVO=true|dGV\ chromatin=false)}$. Final risk ratios and their

confidence intervals are reported as the median and 95% confidence interval from an empirical distribution of risk ratio estimates. The empirical distributions were created by randomly resampling the data set with replacement and recreating 1000 independent networks and their risk estimates. A correction factor ($k = 1/N$) or a network smoothing value (0.01) was used to prevent zero-state probability estimates during bootstrapping.

## Reporting summary

Further information on research design is available in the Nature Portfolio Reporting Summary linked to this article.

## Data availability

The final classification and network data matrices used for CHD phenotype classification and Bayesian network analyses are provided in the Supplementary Data file. The classification training data matrix includes 698 phenotypes for 3,000 CHD patients encoded as binary variables from Fyler phenotype codes. The network data matrix encodes genetic variants (dGVs identified by GEM), surgical outcomes variables, and phenotypic variables used for constructing the Bayesian networks shown in the paper. All damaging variants and genotypes identified by GEM and used for risk prediction analyses are listed in the Supplemental Data file. Exome sequencing data used in this study was generated through an ongoing effort by the National Heart, Lung, and Blood Institute's (NHLBI) Bench to Bassinet (B2B) project to sequence children with CHD. Sequence data are deposited in the database of Genotypes and Phenotypes (dbGaP) under accession numbers phs000571.v1.p1, phs000571.v2.p1 and phs000571.v3.p2 and are available, under controlled access, to other investigators by request [https://www.ncbi.nlm.nih.gov/projects/gap/cgi-bin/about.html#request-controlled/]. Exome sequence and phenotype data, including phenotype variables examined but not used in the final networks, are also available, with controlled access [https://b2bperms.research.cchmc.org/request/], to investigators through the PCGC's HeartsMart database [https://heartsmart.pcgcid.org/].

## Code availability

Open source and published software packages are listed in the Methods. GEM was used to identify damaging variants from exome sequence data. GEM is a commercial tool for AI-assisted clinical interpretation of WES and WGS. It has been licensed by University of Utah Hospitals from Fabric Genomics Inc. and is used by Utah faculty, staff, and affiliates for WGS analyses campus wide. Additional licensing information is available from Fabric Genomics Inc. [https://fabricgenomics.com/].

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

## Acknowledgements

We thank the patients and their families for participating in the Pediatric Cardiac Genomics Consortium and Bench to Bassinet research programs. This research would not be possible without the clinical professionals involved in patient recruitment, and we thank those at the following institutions including the Columbia Medical School: D. Awad, C. Breton, K. Celia, C. Duarte, D. Etwaru, N. Fishman, E. Griffin, M. Kaspakoval, J. Kline, R. Korsin, A. Lanz, E. Marquez, D. Queen, A. Rodriguez, J. Rose, J.K. Sond, D. Warburton, A. Wilpers, and R. Yee; (2) the Children's Hospital of Los Angeles: J. Ellashek and N. Tran; (3) the Children's Hospital of Philadelphia: S. Edman, J. Garbarini, J. Tusi, and S. Woyciechowski; (4) the Harvard Medical School: J. Geva and M. Borensztein; (5) the Icahn School of Medicine at Mount Sinai: A. Julian, M. Mac Neal, Y. Mendez, T. Mendiz-Ramdeen, and C. Mintz; (6) the University College London: B. McDonough, K. Flack, L. Panesar, and N. Taylor; (7) the University of Rochester School of Medicine and Dentistry: E. Taillie; and (8) the Yale School of Medicine: N. Cross. We thank Nick Felicelli, Prakash Velayutham, and the staff at the Cincinnati Children's Hospital Medical Center for computational support and for providing access to the HeartsMart database. We thank Carson Holt, Shawn Rynearson, Barry Moore at the Utah Center for Genomic Discovery and the staff at the Utah Center for High Performance Computing for high-throughput processing of patient sequence data. The Pediatric Cardiac Genomics Consortium (PCGC) program is funded by the National Heart, Lung, and Blood Institute, National Institutes of Health, U.S. Department of Health and Human Services through grants UM1HL128711 (M.T.F., M.Y., H.J.Y.), UM1HL098162 (M.B.), UM1HL098147 (J.W.N., A.E.R., C.E.S.), UM1HL098123 (B.D.G.), UM1HL128761 (D.B.), U01-HL098153 (E.G.), U01-HL098163 (W.K.C.), U01HL131003 (M.W.) and HL162356 (C.E.S.). This manuscript was prepared in collaboration with investigators of the PCGC and has been reviewed and/or approved by the PCGC. PCGC investigators are listed at [https://benchtobassinet.com/?page_id=119].

## Author contributions

W.S.W., E.J.H., M.Y., and M.T.F. conceived and planned the project; M.T.F. and W.S.W. analyzed the data and created the manuscript; E.J.H., R.Z., advised the network analysis and presentation; T.A.M. provided clinical interpretation and analysis support; E.F. contributed to the GEM implementation; M.T.F. and M.Y. oversaw all aspects of the network and statistical analyses; M.W. provided network access to data; W.S.W., E.J.H., T.A.M., R.Z., D.B., M.T.B., M.B., W.K.C., J.W.G., B.D.G., E.G., P.G., J.W.N., A.E.R., S.U.M., J.E.M., C.E.S., J.G.S., Y.S., H.J.Y., N.R.B., E.R.G., M.T.B, M.Y., and M.T.F. edited the final manuscript.

## Competing interests

The authors declare the following competing interests: M.Y. – GEM commercialization through Fabric Genomics, Inc; E.F. is an employee of Fabric Genomics. The remaining authors declare no competing interests.

## Additional information

¹Department of Human Genetics, University of Utah, Salt Lake City, UT 84112, USA. ²Department of Biomedical Informatics, University of Utah, Salt Lake City, UT 84108, USA. ³Department of Pediatrics, Maine Medical Center, Portland, ME, USA. ⁴Department of Obstetrics and Gynecology, University of Utah, Salt Lake City, UT 84112, USA. ⁵Pediatric Cardiothoracic Surgery, University of Utah, Salt Lake City, UT, USA. ⁶Fabric Genomics Inc, Oakland, CA 94612, USA. ⁷Department of Pediatrics, Stanford University School of Medicine, Stanford, CA 94305, USA. ⁸Department of Surgery, University of California, San Francisco,

CA 94143, USA. ⁹Department of Pediatrics, Yale University School of Medicine, New Haven, CT 06510, USA. ¹⁰Department of Genetics, Yale University School of Medicine, New Haven, CT 06510, USA. ¹¹Department of Pediatrics, Boston Children's Hospital, Harvard Medical School, Boston, MA 02115, USA. ¹²Division of Cardiothoracic Surgery, Department of Surgery, Children's Hospital of Philadelphia, and the Perelman School of Medicine, University of Pennsylvania, Philadelphia, PA, USA. ¹³Mindich Child Health and Development Institute and Departments of Pediatrics and Genetics & Genomic Sciences, Icahn School of Medicine at Mount Sinai, New York, NY 10029, USA. ¹⁴Division of Cardiology, Children's Hospital of Philadelphia, Department of Pediatrics, The Perelman School of Medicine, University of Pennsylvania, Philadelphia, PA, USA. ¹⁵Department of Surgery, Yale University, New Haven, CT, USA. ¹⁶Department of Cardiology, Boston Children's Hospital, and Department of Pediatrics, Harvard Medical School, Boston, MA 02115, USA. ¹⁷Division of Newborn Medicine, Department of Medicine, Boston Children's Hospital, Boston, MA, USA. ¹⁸Department of Cardiovascular Surgery, Boston Children's Hospital, Harvard Medical School, Boston, MA 02115, USA. ¹⁹Departments of Genetics and Medicine, Harvard Medical School, Boston, MA 02115, USA. ²⁰Cardiovascular Division, Massachusetts General Brigham, Boston, MA 02115, USA. ²¹Departments of Systems Biology and Biomedical Informatics, Columbia University, New York, NY 10032, USA. ²²Division of Biomedical Informatics, Department of Pediatrics, Cincinnati Children's Hospital Medical Center, Cincinnati, OH, USA. ²³Molecular Medicine Program, University of Utah, Salt Lake City, UT 84112, USA. ²⁴Nora Eccles Harrison Cardiovascular Research and Training Institute, and Division of Pediatric Cardiology, Salt Lake City, UT 84108, USA. ✉e-mail: myandell@genetics.utah.edu; Martin.Tristani@utah.edu

