## [Transparent Peer Review File · Nature Communications]

Genome Sequencing is Critical for Forecasting Outcomes following Congenital Cardiac Surgery

Corresponding Author: Dr Martin Tristani-Firouzi

Version 0:

Reviewer comments:

Reviewer #1

(Remarks to the Author)

In this manuscript Watkins et al. present an analysis of the subset of the PGC cohort of patients with congenital heart defects in order to predict post-surgical clinical outcomes based on genotypic and phenotypic variables. As the authors state, this is indeed a very heterogeneous cohort (i.e. many different cardiac phenotypes, and the same cardiac phenotype being caused by different genetic etiologies), making it hard to build good predictive models. Even if such models would be able to accurately predict surgical outcomes, it's unclear how that would translate to the clinic as most CHD-related surgeries happen before any genetic assays are performed. It's not very clear who the intended audience of this paper would be, as no new insights into CHD genetics or biology are presented, and clinical outcome predictions would require additional evidence to be actually actionable. I also have several concerns about the presented manuscript:

- My main concern is with the robustness of the findings. Given the many subsets/conditions of the tests (i.e. LVO patients carrying de novo mutations in chromatin modifying genes), it's very unclear from the presented results how many data points of each conditional state there are to make these claims. Ideally, these findings would be validated (preferably in an independent validation cohort) in order to instill confidence in these findings (as the authors state in the discussion). At the very least the number of data points backing up the relative risks presented in the figures should be shown as conditional probability distributions with low number of datapoints can be highly unreliable. Also in the feature selection paragraph (page 9, line 218), it's unclear what variables are exactly considered, a list of all the variables, their type and their respective distributions and/or the number of positive cases for that variable would improve transparency.

- Bayesian networks make the assumption that nodes are conditionally independent of its non-descendants given its parents, it would be good for the authors to show that this is indeed the case and if not, that they discuss how this might affect the results. It seems highly unlikely that strongly correlated phenotypes and their genetic components such as those found in CHD would be independent.

- A key aspect of the methodology used to obtain the results relies on the use of commercial software (Fabric GEM), for which the code is not available and is as such difficult to reproduce for the scientific community. This seems somewhat at odds with the reporting standards and code availability policies of Nature Portfolio journals, but I leave that to the discretion of the editorial team.

- When discussing the phenotype automated classification performance on page 7 line 169, it would be more suitable to show the classification metrics on proper testing/validation sets.

- Is the manual categorization of the phenotypes into the distinct categories necessary? Can't the AI model learn the structure in the data without having to do such extensive feature engineering?

- I don't understand how the absence of damaging genotypes can be considered protective for adverse clinical outcomes, while the presence of said variants increases the risk. Relative to what is this absence of variants protective?

Minor:

The authors state that they are underpowered in several of the disease categories. Can the authors clarify what would be a adequately powered sample size to detect effect sizes similar to the ones found in LVO/HTX?

Reviewer #2

(Remarks to the Author)

This manuscript by Tristani and colleagues describes the use of Fabric Genomics AI algorithm (Fabric GEM) to prioritize sequence variants from WES data of PCGC CHD patient cohort and identifying their association with congenital heart defect phenotypes classified using the Boston Fyler code and further translated into HPO terms. This led to the identification of what the authors refer to as Gene Pathway that are associated with four broad category of CHD comprising atrioventricular canal defects (AVC), conotruncal defects (CTD), heterotaxy (HTX), Left ventricular outflow tract (LVO) lesions, and also a fifth category, OTH (others) that is a catchall for everything else, such as ASD, VSD, and other complex lesions not captured by the four CHD categories. Analysis of risk for CHD phenotypes by gene pathway showed significant associations for HTX, LVO, and surprisingly also OTH, while for AVC and CTD, it is said that the analysis was underpowered. Since OTH is a mixture of CHD phenotypes, it is surprising that chromatin genes emerged as a gene pathway – what does this mean for specificity of the phenotyping-gene/pathway assignment? Then further combining the results of this Fabric Genomics analysis with clinical outcome data for these same patients from the STS database, they used Bayesian network analysis to determine risk for different adverse clinical outcomes in the context of the CHD phenotypes and genes/gene pathways, hence explaining the title of the manuscript: Genome Sequencing is Critical for Forecasting Outcomes following Congenital Cardiac Surgery. Given the authors conclusion that patient genomic data can be used for forecasting outcome after congenital cardiac surgery, it is striking that no substantive discussion was provided on how risk assessment of surgical outcomes can either change clinical care from current practice, or help improve patient outcome. This should be discussed to make credible the clinical value of forecasting outcomes for CHD patients..

Overall, this manuscript will be of interest to the audience of Nature Communication at large, but there are some details of the analysis not provided that should be included to confirm rigor of the analysis and also ensure others can replicate these findings in the future. Also, some of the unexpected results should be explained/discussed. Below I provide specific comments/queries for the authors' to consider and address as appropriate:

1. The use of Fabric GEM led to the results shown in Figure 1. I find these result interesting, but the paucity of details of the actual analysis carried out makes it difficult to fully interpret the results and its significance.

-Table 1 should include total number of patients included in each phenotype category with (n=XX) added under the phenotype column heading. Currently, the number of patients with damaging variants in the different gene pathways are shown in the column labeled as "n", but no information is provided on how many patients fall within the different phenotype classification.

- If a patient has both an AVSD and also has CTD with outflow tract malalignment defect, is it included as having AVC or CTD? If each patient is assigned to only a single defect classification, how is this prioritized? Brief discussion of these important points should be included in the Methods

-For the gene pathways, are some patients represented in multiple gene pathways, or is each patient prioritized with assignment to only a single gene pathway?

-For gene pathways, how many and what specific genes are included, and how were these gene lists generated? What is the CHD gene list used? Is there overlap of the CHD gene list with the other gene list? If so, how is this redundancy accounted for? Is there need for multiple test correction to account for the number of genes included in the different pathways? Such information will allow others to replicate these findings in the future and also help the reader to better understand some of the conclusions such as:

We were underpowered to detect enrichment in damaging genotypes in the AVC and CTD phenotype classes. It would be useful to know how many subjects had AVC and CTD vs. HTX and LVO.

The authors also stated, "Notably, damaging genotypes in these pathways were not enriched in HLHS patients, further underscoring the complex genetic landscape underlying CHD". How man HLHS patients were included, presumably in the LVO category? Is it simply underpowered rather than a reflection of complex genetics?

-For FoxJ1 pathway, since there are only 6 patients identified with pathogenic variants. Given the risk ratio reported in the table, it is likely there are three with HTX, and one each of the remainder three in CTD, LVO, and OTH. Is this the case? Since this is a recessive genetic model, can the authors please provide the genes and the biallelic variants recovered. Ditto for the Cilia genes, which also falls under recessive genetic model. How many patients does it represent of the 35 identified with cilia variants? What are the genes recovered and the biallelic variants recovered? Please provide a table or figure showing the biallelic variants. What is the extent overlap of cilia gene list vs. the FoxJ1 genes? Thus are some of the same patients driving the results of FoxJ1 also driving the results seen with the Cilia genes? I noted for both the Cilia genes and FoxJ1 genes, only HTX phenotype comes up with significant increased risk, which would suggest that the cilia gene list contribution may be largely accounted for by the FoxJ1 regulated cilia genes? This is where providing the actual genes and biallelic variants recovered in the FoxJ1 vs. Cilia genes in the HTX patients might be helpful.

-Given the previous report from a large scale mouse screen by Lo et al of an enrichment for cilia genes, were there any

correlation of the damaging variants identified in the cilia genes in the patients, and the cilia genes recovered in their mouse screen? It is interesting to note that the authors stated, "Biallelic damaging genotypes were observed in multiple patients for several genes including DYNC2H1 (3), DNAH5 (3), LAMA2 (3), GDF1 (2), and IFT140 (2). All three of the cilia genes - Dync2h1, Dnah5, and Ift140, were in fact recovered in the Lo CHD mouse screen. They also recovered a mutation in FoxJ1. Given the mouse screen was designed to recover recessive mutations, their mouse results fit well with the recovery of biallelic damaging genotypes, supporting a recessive model of disease.

-The finding of significance for CHD genes in the LVO and OTH class is puzzling. How many genes are in the CHD gene list and does it include genes in the other lists? Is the finding a reflection of the very large number of genes in this list and perhaps larger number of LVO and OTH patients included in this analysis? As OTH is a catchall of simple and complex lesions, it would be important to know what are the patient phenotypes that contribute to the significant risk findings – are these the patients with very complex lesions not easily classified into any one of the four lesion types used in this analysis or do they also include patients with isolated VSD/ASDs?. Ditto regarding the increased risk seen with CHD genes – what are the specific phenotypes and genes recovered? Does the CHD gene list also include the chromatin genes, ie. might the chromatin genes drive the results seen in the CHD genes? The gene lists should be provided and more clarification of the results are needed to allow the reader to understand the significance of these findings.

2. In the clinical outcome analysis, data from the STS surgical database was used to track postsurgical outcome in these same patients analyzed for pathogenic variants. The Bayesian network analysis yielded some interesting association of increased risk for postsurgical complications comprising mortality, cardiac arrest and requirement for mechanical ventilation postsurgically.

-Please provide number of patients included in each of the risk analysis in Figure 1c, 2, 3. ie. please include n=XX for each outcome risk analysis shown in these forest plots. Please provide p values. The discussion for this data was focused on cardiac arrest for HTX patients, but I noted VENT outcomes for HTX patients yielded the highest risk, higher than for cardiac arrest. Given the finding of cilia and Foxj1 gene pathway enrichment for HTX patients, some discussion of the relevance of these findings to the increased adverse VENT outcomes should at least be mentioned.

In Figure 2, similar analysis was conducted for extracardiac anomalies. It would be useful to know what are the ECA recovered in the subjects with the pathogenic variants, and are there recurrent defects, such as limb anomalies, cleft palate, or renal defects. Could the authors comment on why there is such increase risk seen with ECA? Is this related to the ECA or simply that the complexity of birth defects reflects the central importance of the gene in multiple development pathways?

-The presumed value of having genomic data and being able to assess risk for adverse outcomes following congenital cardiac surgery is to help inform postsurgical management of CHD patient care, thereby helping to improve postsurgical outcome. Please discuss how risk assessment based on having genomic data can improve postsurgical outcome for CHD patients. A brief consideration of how such knowledge might change clinical care needs to be addressed, as otherwise what is the point of knowing these risks?

Reviewer #3

(Remarks to the Author)

In the manuscript 'Genome Sequencing is Critical for Forecasting Outcomes following Congenital Cardiac Surgery' the authors present genome sequencing data derived from patients with a congenital heart defect and show how genome sequencing has the potential to be used as risk stratification for children with congenital heart defects.

I would like to congratulate the authors for this very interesting manuscript which highlights the potential of genetic data in risk stratification of children with a congenital heart defect (CHD). Although this is an original and high-quality manuscript, I believe it could be improved in several ways.

1. Data may be presented in a more clear way:

- Although the authors describe the CHD categories in an appropriate way, it is unclear why some patients have been classified in a certain category. For examples in supplemental table 8 in the AVC category, beside classic AVC pathology such as AVSD, there are patients with coarctation, aortic valve anomalies, transposition of the great arteries or truncus. Equally, some diagnosis seem to be classified into different categories (Ex: pulmonary valve stenosis or VSD appear in all categories). Could the authors explain.

- Supplemental tables 9 and 10 could benefit from having some clinical data added. I would suggest to add the underlying heart defect and category assigned.

- In supplemental table 10, it is unclear whether the variants found, were found in homozygosity or heterozygosity for all patients. This is important to clearly mention, given that carrier state alone would probably not explain the CHD.

- I could not found the variables included for surgical outcome. In supplemental table 2, only mortality has been added. I might have missed it.

2. Many variants have a VUS classification in ClinVar, how sure are the authors that these variants are disease causing? Has there any validation been performed to compare the results of GEM in relation to the ACMG classification?

3. In the same line, how sure are the authors that all found variants are causing the CHD phenotype. For example, a MYBPC3 variant in the context of CHD, could be an incidental finding depending on the underlying CHD.

4. It's remarkable that no associations have been found in patients with tetralogy of Fallot and AVSD. These categories of CHD have always been considered to have the largest genetic basis. Could the authors comment?

5. Around 10% of CHD patients carried a (likely) pathogenic variant which could explain the phenotype. Although this is indeed in line with the published literature, it leaves the great majority (90%) of cases unexplained. It could be interesting

that the authors add a word on future directions to solve these cases.

Version 1:

Reviewer comments:

Reviewer #1

(Remarks to the Author)

Although I was initially highly critical of the presented manuscript, the replies of the authors have addressed my concerns. I've been convinced by the arguments made about the potential impact of the manuscript as a pioneering paper in the use of WGS for pre/post-surgical outcome prediction. How large the audience for this will be, and if this will indeed affect clinical practice, is difficult to predict. However, we will never know unless these type of works are published and validated in future works.

I also commend the authors on the clarity of their retorts. Highly technical concepts are very well formulated in a relatively easy manner to understand, even for non-statisticians/data analysts. I believe this will widen the scope of readership significantly.

I have no further concerns/remarks.

Reviewer #3

(Remarks to the Author)

I would like to thank the authors for their efforts reviewing the manuscript.

I believe it has improved considerably and that several issues are now much more clear.

I have just one minor remark:

Although I believe that GEM as a system has been properly validated, I keep struggling with the large amount of VUS (based on ClinVar reporting) included. Looking at supplemental table 1, it seems that VUS get a lower GEM score than (likely) pathogenic variants. I would appreciate that the authors add a comment on this in their discussion.

RESPONSE TO REVIEWER COMMENTS

(Reviewer comments are italicized)

Reviewer #1 (Remarks to the Author):

In this manuscript Watkins et al. present an analysis of the subset of the PCGC cohort of patients with congenital heart defects in order to predict post-surgical clinical outcomes based on genotypic and phenotypic variables. As the authors state, this is indeed a very heterogeneous cohort (i.e. many different cardiac phenotypes, and the same cardiac phenotype being caused by different genetic etiologies), making it hard to build good predictive models. Even if such models would be able to accurately predict surgical outcomes, it's unclear how that would translate to the clinic as most CHD-related surgeries happen before any genetic assays are performed.

Rapid WGS has emerged as the standard of care for critically ill newborns, including those in the cardiac ICU. Moreover, the rapid decline in sequencing costs makes a genome more attractive than the outdated previous standard genetic test for children with CHD, i.e., chromosomal microarray (CMA). Moving the needle on insurance coverage for WGS requires published data showing the value of such testing. By confirming that genomic information in the context of specific clinical variables is predictive of adverse post-surgical outcomes, our manuscript provides further evidence of the value of genomics to insurance companies across the US. With respect to timing of surgical intervention, many children with CHD undergo cardiac surgery outside of the newborn period, allowing sufficient time for genome sequencing to be completed pre-operatively. As turn-around times shorten and costs decline for ultra-rapid and rapid WGS, genomic data are increasingly available in the pre-operative period to allow for more personalized risk assessments and more tailored, early interventions to mitigate adverse post-operative outcomes.

It's not very clear who the intended audience of this paper would be, as no new insights into CHD genetics or biology are presented, and clinical outcome predictions would require additional evidence to be actually actionable.

Our paper is the first to show the predictive value of genomics beyond diagnostics alone, in children with a complex disorder, such as CHD. Our results are predicated on explainable AI approaches that are relevant to broad medical disciplines. As such, we believe that researchers and clinicians interested in precision medicine, bioinformatics and the intersection of computational approaches with large clinical/genomic datasets will be interested in this manuscript. Our

presentations of these findings at national/international scientific sessions have generated tremendous enthusiasm, resonating with a large cadre of precision medicine proponents.

I also have several concerns about the presented manuscript:

- My main concern is with the robustness of the findings. Given the many subsets/conditions of the tests (i.e. LVO patients carrying de novo mutations in chromatin modifying genes), it's very unclear from the presented results how many data points of each conditional state there are to make these claims.

No doubt larger (future) cohorts, especially inception cohorts, will help to test and refine our risk predictions, especially for the rarer cases, but we emphasize that Bayesian Networks are the gold-standard for calculating the probabilities of rare events. This is because they allow for the incorporation of prior knowledge, explicitly model uncertainty, and provide established best-practice methodologies to avoid overfitting. Moreover, Bayesian methods model uncertainty using a probability distribution (the posterior distribution) for each parameter rather than a single point estimate. This last feature is particularly valuable in low-data scenarios, allowing the model to express uncertainty through conditional probability distributions even with limited data. Moreover, the large size of the PCGC cohort means that resampling can be used to conveniently document uncertainty. The confidence intervals we provide for our risk estimates were generated by simultaneously resampling the data and rebuilding the Bayesian network each iteration. In general, wide confidence intervals reflect low case numbers, telegraphing this fact to the reader. We acknowledge that risk projections for some of the rare, high-risk scenarios are rough estimates, but there are no sharper tools or larger datasets. Future analyses using even larger cohorts will one day refine these estimates. Our manuscript provides a necessary starting point for that future work. We have broadened our discussion of these points in the revised manuscript.

New language in the Discussion:

“Overall, the number of adverse events in probands with damaging genotypes in some clinical contexts was relatively low, despite an initial corpus of >2000 cases. It is well established that the Bayesian statistical framework is particularly well-suited for predictions when numbers are limiting¹; indeed, this was our motivation for employing this framework. Bayesian approaches allow for the incorporation of prior knowledge, explicitly model uncertainty, and provide established best-practice methodologies to avoid overfitting. Moreover, Bayesian methods model uncertainty using a probability distribution (the posterior distribution) for each parameter rather than a single point estimate. This last feature is particularly valuable in low-data scenarios, allowing the model to express uncertainty through conditional probability distributions even with

limited data. This feature is particularly valuable in low-data scenarios, allowing the model to express uncertainty through conditional probability distributions even with limited data². The confidence intervals we report were generated by simultaneously resampling the data and rebuilding the net. This is a gold standard approach for estimating uncertainty when data are limiting. In general, the wide confidence intervals reflect this data scarcity. Taken together, these features allow Bayesian networks to make better informed predictions in low-data scenarios than do traditional frequentist approaches¹⁻³. It should be born in mind that risk projections for some of the rare, high-risk scenarios are rough estimates. No doubt future analyses using even larger cohorts will one day refine these estimates. Our results provide a necessary starting point for that future work”.

At the very least the number of data points backing up the relative risks presented in the figures should be shown as conditional probability distributions with low number of datapoints can be highly unreliable.

We now include the counts for each risk predictions shown in Figures 1c, 2, and 3 in Supplemental Table 4 and describe the counts in Results and Figure Legends. Again, the wide confidence intervals for these rare events reflect this data scarcity and serve to telegraph this fact to the reader. We discuss the implications in the Results section (described below).

“The patient counts for the relative risk ratios presented here range from 1 to 179 (Supplemental Table 4). While the number of adverse events in some of these genetic and clinical contexts was relatively low, the Bayesian statistical framework is particularly well-suited for predictions when numbers are limiting^{4,5}, which was our motivation for employing this framework.”

Ideally, these findings would be validated (preferably in an independent validation cohort) in order to instill confidence in these findings (as the authors state in the discussion).

We agree that replication in an independent, external cohort is important for reproducibility and scientific rigor. However, replication is not possible or meaningful, as no other CHD cohorts in the world have the same depth or breadth of post-surgical variables with genomic information. The lack of comparable cohorts reflects the current constraints of the field and underscores the unique nature of the PCGC dataset.

While replication is not feasible today, many relationships captured by the Bayesian Network model are biologically and clinically plausible, and importantly, align with existing literature. For example, the risk of prolonged mechanical ventilation in patients with damaging cilia genotypes in the context of HTX and STAT4 surgeries is supported by literature linking cilia dysfunction and HTX with post-operative respiratory complications⁶⁻⁹. The rare cases are of particular interest from a clinical perspective, as their rarity makes the additional clinical guidance provide by sequencing

data all the more valuable. Moreover, we believe there are reasons these cases are rare. It should be kept in mind that the PCGC cohort is not an inception cohort, and such cases may be dying prior to study inclusion, a hypothesis consistent with their high risks of morbidity and mortality as predicted by the Bayesian Networks. These considerations reinforce our conclusion that uniform sequencing of all CHD newborns, as early as possible, is essential for improving the standard of care, especially for these rare cases. We now discuss these points in more detail in the revised manuscript.

New language in the Discussion limitations paragraph:

“Finally, we acknowledge that replication in an independent, external cohort is important for reproducibility and scientific rigor. However, replication is not possible or directly comparable, as no other CHD cohorts in the world have the same depth or breadth of post-surgical variables with genomic information. The lack of comparable cohorts reflects the current constraints of the field and underscores the unique nature of the PCGC dataset. While replication in an independent cohort is not currently feasible, the relationships captured by the Bayesian network model are biologically and clinically plausible, and importantly, align with existing literature. For example, the risk of prolonged mechanical ventilation in patients with damaging cilia genotypes in the context of HTX and STAT4 surgeries is well-supported by literature linking cilia dysfunction and HTX with post-operative respiratory complications. As more data become available in external CHD cohorts, we plan to validate our model prospectively.”

Also in the feature selection paragraph (page 9, line 218), it's unclear what variables are exactly considered, a list of all the variables, their type and their respective distributions and/or the number of positive cases for that variable would improve transparency.

We now include Supplemental Table 12 that describes the frequencies and patient counts for all STS surgical variables screened and included or excluded from the analyses.

- Bayesian networks make the assumption that nodes are conditionally independent of its non-descendants given its parents, it would be good for the authors to show that this is indeed the case and if not, that they discuss how this might affect the results. It seems highly unlikely that strongly correlated phenotypes and their genetic components such as those found in CHD would be independent.

We agree that highly correlated pairs or correlated sets of variables can be problematic in Bayesian network analyses. However, this is not the case for variables used in our networks. We have now added a new section called “Bayesian networks and variable correlations” to the Supplementary Data providing full details regarding the correlation among the variables used in the Bayesian networks.

“We note that in the Bayesian networks, the assumption of non-independence between the parents and non-descendant nodes applies to the learning and structure of the Bayesian network. This assumption does not imply that variables that are conditionally independent in the network cannot be correlated. Conversely, two apparently non-correlated variables are not necessarily independent once they are examined in network context because one or more other network variables may reveal a new conditional dependency.”

Indeed, the ability of Bayesian networks to capture events such as these and to deliver the correct posterior probably is a major motivation for our using them.

We now provide Supplementary Figures 4a, b showing the all pairwise correlations and P-values among the variables used Figures 1a, b in the main text. Pairwise correlations among these variables range from -0.1 to 0.3 except for the expected high correlation for LVO and HLHS.

We have also added the following sentences to the Methods section:

“Because highly correlated variables may influence Bayesian network structure learning and risk estimation, variables included in the networks were screened for colinear and multicollinear states by correlation analysis (Supplemental Figures 5a and 5b). Network variables were not colinear and pairwise correlations were between -0.11 and 0.30, except for LVO and HLHS.”

We have also added new material discussing the impact of the sample size on the networks to the Supplementary Data and the Methods Section.

“The accuracy of network inferred risk estimates can be affected by sample size. Many studies advocate Bayesian analysis for low sample size applications. With low sample sizes, accurate posterior probabilities and risk estimates are highly dependent on informative and unbiased prior probability estimates. Our initial probability estimates for variables used in the study are based on 2,253 CHD patients from multiple surgical sites and are, therefore, likely to be representative probability estimates for a critical CHD surgical cohort and without strong biases.”

- A key aspect of the methodology used to obtain the results relies on the use of commercial software (Fabric GEM), for which the code is not available and is as such difficult to reproduce for the scientific community. This seems somewhat at odds with the reporting standards and code availability policies of Nature Portfolio journals, but I leave that to the discretion of the editorial team.

We appreciate the reviewer’s concern regarding the availability of software for reproducibility by the scientific community. The use and effectiveness of the Fabric-GEM software package is now established and documented by several publications¹⁰⁻¹⁴. The software is currently used by Rady Children’s Hospital, The University of Utah Hospitals, The Broad Institute, and numerous other institutions for analysis of human exome and whole genome sequencing data. Other commercial software packages for bioinformatics, engineering, and biological sciences — such as Qiagen’s

Ingenuity software, Illumina’s Dragen Variant caller, and Golden Helix’s software products — are widely used in scientific publications without direct source code availability. To provide readers the information needed to obtain and evaluate Fabric-GEM, we added the following information to the Data and software availability section:

“GEM is a commercial tool for AI-assisted clinical interpretation of WES and WGS. It has been licensed by University of Utah Hospitals from Fabric Genomics Inc. and is used by Utah faculty, staff, and affiliates for WGS analyses campus wide. Additional licensing information is available from Fabric Genomics Inc.”

We have added four additional citations that use and evaluate Fabric-GEM to the manuscript.

- When discussing the phenotype automated classification performance on page 7 line 169, it would be more suitable to show the classification metrics on proper testing/validation sets.

We utilized a five-fold cross-validation approach for the supervised learning of the categories and applied conservative practices such as 50% subsampling and low learning rates to reduce overfitting the classification model. We agree that an independent validation cohort could be used to benchmark the classifier performance, but none exists. We now acknowledge this limitation and have added a cautionary note in the Supplemental Methods,

“We did not have access to an independent manually curated validation cohort which would have provided more rigorous performance metrics. Due to possible differences among physicians and sites in assigning Fyler terms, the classifier accuracy should be further evaluated on other data sets as they become available.”

- Is the manual categorization of the phenotypes into the distinct categories necessary? Can’t the AI model learn the structure in the data without having to do such extensive feature engineering?

We initially tried unsupervised clustering approaches (e.g., PCA) but found that phenotype clusters were somewhat diffuse and overlapping likely due to equal weighting of the Fyler terms in this approach. The PCGC had previously developed the five phenotype categories based on biological patterns observed in normal heart development. We decided to use a gradient-boosted decision tree (XG-Boost; see xgboost.readthedocs.io for details) to first learn the patterns and information gain for CHD features (Fyler terms) developed by expert cardiologists and then used to assign a patient to a phenotype class. XG-Boost is a supervised learning method that has demonstrated excellent performance in many classification competitions. This method produced better specificity for individual patients and provided a probabilistic evaluation for the assignment of each patient to a single phenotype class. We now acknowledge the many challenges for CHD phenotype

classification in general and add additional information about the classification method to the Methods section.

“Considerable heterogeneity exists among patients’ heart phenotypes and the five phenotype categories. For example, pulmonary stenosis and ventricular septal defects occur in all five phenotype categories. The observed rate of each Fyler code for each phenotype category in the 2,253 CHD analysis individuals is shown in Supplemental Table 7. This heterogeneous data structure suggested that a supervised learning model that leverages prior physician classification information¹⁵ and weights each cardiac defect based on expert knowledge would perform better than an unsupervised classification approach. Therefore, we elected to use a gradient-boosted decision tree model trained on physician-based classification of 3,000 patients to learn the importance or information gain for each Fyler code. Once trained, the algorithm can probabilistically assign each proband to one of the five phenotype categories. In the final algorithm, Fyler codes strongly associated with a specific phenotype category in the training data greatly increase the probability that the patient belongs to a specific category. For example, tetralogy of Fallot is nearly always assigned to CTD, while hypoplastic left heart is highly predictive of LVO. In contrast, pulmonary stenosis has lower predictive value for classification.”

- I don't understand how the absence of damaging genotypes can be considered protective for adverse clinical outcomes, while the presence of said variants increases the risk. Relative to what is this absence of variants protective?

We agree that our use of the word “protective” is confusing. We have replaced protective with “is associated with” to indicate that those patients without a damaging variant have lower risk as compared to patients with damaging variants.

Minor:

The authors state that they are underpowered in several of the disease categories. Can the authors clarify what would be a adequately powered sample size to detect effect sizes similar to the ones found in LVO/HTX?

LVO/HTX comprise the most complex forms of CHD and are associated with the greatest morbidity and mortality. The number of probands experiencing adverse outcomes in the AVC, CTD, and OTHER categories and harboring damaging gene pathway variants was too low for reliable risk prediction. The reasons for this are multi-factorial, including genetic and phenotypic heterogeneity, low number of patients in some categories (e.g., AVC=64 probands) as well as excellent surgical outcomes in these categories. For example, no patient with AVC died post-

operatively in this cohort. Assuming a baseline AVC mortality of ~1%, if we were to consider a 2-fold increase in mortality in AVC patients with a damaging cilia genotype, we would need 960 AVC patients, based on a simple analysis for 0.8 power and alpha 0.05.

We edited the Results Section with new language (in italics):

“The number of probands experiencing adverse outcomes in the AVC, CTD, and OTHER categories and harboring damaging gene pathway variants was too low to warrant generation of Bayesian networks for outcomes prediction in these CHD phenotypes. *The reasons for this are multi-factorial, including genetic and phenotypic heterogeneity, low number of patients in some categories, as well as excellent surgical outcomes in these categories. For example, no patient with AVC died post-operatively in this cohort.* Consequently, larger cohorts are necessary to adequately predict the impact of genetics on outcomes for these CHD phenotypes.”

Reviewer #2 (Remarks to the Author):

Given the authors conclusion that patient genomic data can be used for forecasting outcome after congenital cardiac surgery, it is striking that no substantive discussion was provided on how risk assessment of surgical outcomes can either change clinical care from current practice, or help improve patient outcome. This should be discussed to make creditable the clinical value of forecasting outcomes for CHD patients.

We thank the reviewer for this suggestion and added an additional paragraph in the Discussion devoted to implications for clinical care.

“The ability to quantify the risk of adverse outcomes allows for crucial, early deployment of potential therapeutic strategies to mitigate these risks and improve patient outcomes. For example, an extensive body of literature links ciliary dysfunction with respiratory complications, including prolonged ventilation, in the post-operative period^{16-9,16,17} Pre-operative knowledge of a damaging cilia genotype would allow for early institution of aggressive airway clearance therapy, mucolytic therapies, inhaled β 2 agonists and specific ventilation strategies to promote mucociliary clearance¹⁶. Avoiding common anesthetic and analgesic agents that are known to impair mucociliary function may also be beneficial in these patients¹⁷. As turn-around times shorten and costs decline for ultra-rapid and rapid WGS, genomic data will be available in the pre-operative period to allow for risk stratification and early interventions to mitigate adverse post-operative outcomes.”

Overall, this manuscript will be of interest to the audience of Nature Communication at large, but there are some details of the analysis not provided that should be included to confirm rigor of the analysis and also ensure others can replicate these findings in the future. Also, some of the unexpected results should be explained/discussed. Below I provide specific comments/queries for the authors' to consider and address as appropriate:

1. The use of Fabric GEM led to the results shown in Figure 1. I find these result interesting, but the paucity of details of the actual analysis carried out makes it difficult to fully interpret the results and its significance.

We have added additional background information about how GEM works to the Methods section. We have added multiple citations that deployed and benchmarked GEM in clinical settings. We have included information on the GEM software availability in the Data and software availability section.

For clarification regarding the GEM and the networks in Figure 1, we have modified the figure legend so that the reader can quickly understand how each network was constructed and how the GEM results were incorporated in each network.

“Bayesian networks display a best machine-learned relationship among genotypes, phenotypes, and outcomes for 2,253 surgical patients with CHD. Each network node represents a present/absent variable. Damaging genotypes in chromatin-modifying genes (CHRMdGV) or cilia-related genes (CILIAdGV) were identified from the exomes of 2253 CHD patients by GEM (see Methods). Phenotype classes were predicted from Fyler codes using XG-Boost. Surgical outcomes for each patient were obtained from the Society of Thoracic Surgeons national database. Relative risks for selected surgical outcomes were then estimated from each network using network-propagated probabilities.”

-Table 1 should include total number of patients included in each phenotype category with (n=XX) added under the phenotype column heading. Currently, the number of patients with damaging variants in the different gene pathways are shown in the column labeled as “n”, but no information is provided on how many patients fall within the different phenotype classification.

We now indicate the number of patients in each of the five phenotype categories in the column headings of Table 1.

- If a patient has both an AVSD and also has CTD with outflow tract malalignment defect, is it included as having AVC or CTD? If each patient is assigned to only a single defect classification, how is this prioritized? Brief discussion of these important points should be included in the Methods.

Our classification method learns the importance (information gain) of each Fyler term using prior classification performed by expert cardiologists on 3,000 CHD patients. Fyler terms strongly predictive for inclusion in a phenotype category (based on the training data) have higher weights and increase the probability that the patient should be assigned to a category. Thus, a patient with AVSD and tetralogy of Fallot would be classified as CTD, while a patient with AVSD, tetralogy of Fallot and HTX would be classified at HTX, driven by the higher importance of the tetralogy of Fallot and HTX terms, respectively, which have higher information gain as learned by the classifier. Of course, the final classification will depend on all Fyler descriptors for the patient and their relative weights. The classifier output is expressed as a probability for each phenotype category based on all Fyler terms for that patient. Finally, the patient is assigned to the category with the highest probability.

We have now added an additional paragraph to the Methods section to better explain this process. (More details and an example of the probabilistic classification is now included in the Supplemental Data and Methods).

“Considerable heterogeneity exists among patients’ heart phenotypes and the five phenotype categories. For example, pulmonary stenosis and ventricular septal defects occur in all five

phenotype categories. The observed rate of each Fyler code for each phenotype category in the 2,253 CHD analysis individuals is shown in Supplemental Table 7. This heterogeneous data structure suggested that a supervised learning model that leverages prior physician classification information⁵ and weights each cardiac defect based on expert knowledge would perform better than an unsupervised classification approach. Therefore, we elected to use a gradient-boosted decision tree model trained on physician-based classification of 3,000 patients to learn the importance or information gain for each Fyler code. Once trained, the algorithm can probabilistically assign each proband to one of the five phenotype categories. In the final algorithm, Fyler codes strongly associated with a specific phenotype category in the training data greatly increase the probability that the patient belongs to a specific category. For example, tetralogy of Fallot is nearly always assigned to CTD, while hypoplastic left heart is highly predictive of LVO. In contrast, pulmonary stenosis has lower predictive value for classification”.

-For the gene pathways, are some patients represented in multiple gene pathways, or is each patient prioritized with assignment to only a single gene pathway?

Yes, probands with damaging variants may be represented in multiple gene pathways. This contrasts with the phenotype categories which are exclusive.

-For gene pathways, how many and what specific genes are included, and how were these gene lists generated?

Specific genes in each gene list are shown in Supplemental Table 3. The pathway gene lists were generated using the reactome pathway browser [reactome.org]. The CHD gene list was curated from the literature and is not pathway specific. The gene lists have been previously described¹⁸. We now add additional details to the Methods section regarding the gene lists.

For additional clarity, we have now added the gene list membership to each gene reported in Supplementary Tables 1 and 2 (previously Supplemental Tables 9 and 10). We have also added the patient phenotype category to each patient reported in Supplementary Tables 1 and 2.

What is the CHD gene list used? Is there overlap of the CHD gene list with the other gene list?

The CHD gene list was curated from the literature and contains genes reported to be associated with CHD. The CHD list contains 35 genes represented in the chromatin-modifying gene list but does not overlap the cilia gene list. This is now noted in the Methods and Results. A previous publication describing the CHD list is now referenced. In the Methods section we now state that,

“The CHD gene list represents genes from the literature known to directly cause or be associated with CHD and has been previously described¹⁸. The CHD gene list contains 35 genes in the

chromatin list, but no genes from the cilia gene list, and represents a diverse set of genes not specific to a single pathway.”

If so, how is this redundancy accounted for? Is there need for multiple test correction to account for the number of genes included in the different pathways? Such information will allow others to replicate these findings in the future and also help the reader to better understand some of the conclusions such as: We were underpowered to detect enrichment in damaging genotypes in the AVC and CTD phenotype classes.

Although there is some overlap among some gene lists, we do not specifically adjust for redundancy among lists. We use a Bayesian approach to estimate relative risks for a given set of genes. Unlike frequentist-based inferential statistical methods that rely on multiple test corrections for interpretation, Bayesian statistical frameworks rely on probabilistic concepts that are especially well suited for estimating risk with low sample sizes. In this context, Bayesian approaches provide a best estimate of relative risks and associations among genotypes, phenotypes, and outcomes given the data. The statistical support for risk estimates is quantified by resampling to provide a 95% confidence interval for each estimate.

We have also clarified our comments regarding the associations with AVC and CTD in the Results section, “Consistent with previous findings, the AVC category showed an association with cilia genes but did not reach significance due to a low number of patients with AVC. The patients with CTD did not show a significant association with these gene lists, but our data set did not include CNVs which are known to show association with CTD defects and tetralogy of Fallot¹⁹.”

It would be useful to know how many subjects had AVC and CTD vs. HTX and LVO.

Phenotype category assignments were unique to each patient; there were AVC (64), CTD (934), HTX (219) and LVO (647). These counts are now shown in the column labels in Table 2.

The authors also stated, “Notably, damaging genotypes in these pathways were not enriched in HLHS patients, further underscoring the complex genetic landscape underlying CHD”. How many HLHS patients were included, presumably in the LVO category? Is it simply underpowered rather than a reflection of complex genetics?

We apologize for the confusion and have re-worded the sentence for clarity (see below). We are likely underpowered. The risk ratio for HLHS patients and damaging chromatin genotypes was 1.7, but the association was not significantly different from 1.0. With a larger sample size, the CIs might narrow. Also, almost all HLHS patients (287/289) were classified as LVO.

“Notably, damaging genotypes in these pathways were not enriched in HLHS patients, although an association might be detectable with a larger sample size.”

-For FoxJ1 pathway, since there are only 6 patients identified with pathogenic variants. Given the risk ratio reported in the table, it is likely there are three with HTX, and one each of the remainder three in CTD, LVO, and OTH. Is this the case?

The patient counts for the FoxJ1 pathway were four HTX, one CTD, and one LVO.

Since this is a recessive genetic model, can the authors please provide the genes and the biallelic variants recovered. Ditto for the Cilia genes, which also falls under recessive genetic model.

We now provide variants, genotypes, and inheritance for all biallelic variants in Supplemental Table 2. All de novo /dominant variants are now provided in Supplemental Table 1. Because GEM provides an assessment of the *genotype*, all patients listed in Supplemental Table 2 are either homozygous recessive or compound heterozygotes. We now indicate the inheritance for each variant (homozygous recessive / compound heterozygous) in an additional column in Supplemental Table 2.

How many patients does it represent of the 35 identified with cilia variants? What are the genes recovered and the biallelic variants recovered? Please provide a table or figure showing the biallelic variants. What is the extent overlap of cilia gene list vs. the FoxJ1 genes?

Patients with damaging genotypes in *FOXJ1* pathway genes are a subset of the 35 patients. We have added the sentence, “Patients with *FOXJ1* pathway mutations accounted for four of nine HTX patients (44%) in the cilia enrichment subset.”

Forty of the 116 *FOXJ1* genes are represented in the large cilia gene list. We provide all gene lists in Supplementary Table 3. As noted above, the gene list information and phenotype categories for each patient with a damaging genotype are now listed Supplemental Tables 1 and 2.

Thus are some of the same patients driving the results of FoxJ1 also driving the results seen with the Cilia genes? I noted for both the Cilia genes and FoxJ1 genes, only HTX phenotype comes up with significant increased risk, which would suggest that the cilia gene list contribution may be largely accounted for by the FoxJ1 regulated cilia genes? This is where providing the actual genes and biallelic variants recovered in the FoxJ1 vs. Cilia genes in the HTX patients might be helpful.

Yes, patients with damaging variants in the *FOXJ1* genes are contributing to the HTX signal. Twenty other cilia related genes, such as *DNAH5*, *EVC*, *CRB2*, were also identified in the cohort.

We have now updated Supplemental 2 to include the genes, CHD phenotypes, gene list information, variant type, inheritance mode, chromosomal position, etc. for all patients with damaging biallelic variants.

- Given the previous report from a large scale mouse screen by Lo et al [Li et al?] of an enrichment for cilia genes, were there any correlation of the damaging variants identified in the cilia genes in the patients, and the cilia genes recovered in their mouse screen? It is interesting to note that the authors stated, “Biallelic damaging genotypes were observed in multiple patients for several genes including DYNC2H1 (3), DNAH5 (3), LAMA2 (3), GDF1 (2), and IFT140 (2). All three of the cilia genes - Dync2h1, Dnah5, and Ift140, were in fact recovered in the Lo CHD mouse screen. They also recovered a mutation in FoxJ1. Given the mouse screen was designed to recover recessive mutations, their mouse results fit well with the recovery of biallelic damaging genotypes, supporting a recessive model of disease.

We thank the reviewer for the opportunity to further examine the *FOXJ1* pathway genes. All *FOXJ1* pathway genes identified in CHD patients have known associations with primary ciliary dyskinesia and most of the patients' phenotypes are directly supported by the recessive mouse models of CHD reported in Li et. al., (2015). We now list these *FOXJ1* pathway genes in the Results section, “Damaged genes in the *FOXJ1* pathway were *ARMC4*, *CCDC151*, *DNAI1*, *DRC1*, *IFT172*, and *SPEF2*.”

We have added that 4 of the 6 genes cause CHD with heterotaxy in mouse models and cite Li et. al., 2015.

“Patients with *FOXJ1* pathway mutations accounted for four of nine HTX patients (44%) in the cilia enrichment subset. Damaged genes in the *FOXJ1* pathway were *ARMC4*, *CCDC151*, *DNAI1*, *DRC1*, *IFT172*, and *SPEF2*.”

Interestingly, all six genes are linked to cilia dysfunction and to primary ciliary dyskinesia in humans.

-The finding of significance for CHD genes in the LVO and OTH class is puzzling. How many genes are in the CHD gene list and does it include genes in the other lists? Is the finding a reflection of the very large number of genes in this list and perhaps larger number of LVO and OTH patients included in this analysis? Does the CHD gene list also include the chromatin genes, ie. might the chromatin genes drive the results seen in the CHD genes? The gene lists should be provided and more clarification of the results are needed to allow the reader to understand the significance of these findings.

The CHD gene list contains 402 CHD genes identified from the literature and represents a variety of genes, including transcription factors. These genes are all known to directly cause or be associated with CHD. Of the 402 genes in the CHD gene list, 35 overlap the chromatin-modifying gene list and zero overlap the cilia list. Because all genes in the CHD list have known associations with CHD and map to multiple genetic pathways, the CHD gene list more likely to be associated with multiple phenotype categories and heterogeneous CHD phenotypes present in the OTH category.

The gene lists are shown in Supplemental Table 3. The genes, gene list, and specific phenotypes for each reported proband are now available in Supplemental Tables 1 and 2.

Since OTH is a mixture of CHD phenotypes, it is surprising that chromatin genes emerged as a gene pathway – what does this mean for specificity of the phenotyping-gene/pathway assignment?

This may be a result of genetic pleiotropy and too few patients to fully resolve the details of the genotype-phenotype relationship in CHD. We see a signal in the AVC category for damaging cilia genotypes, consistent with the literature, however the broad CIs preclude a definitive association. A larger sample size would likely confirm the association. The observation that CTD category was not associated with the gene pathways/categories is not completely unexpected. CNVs (not studied here) are a known genetic driver in the CTD category, specifically Tetralogy of Fallot.

As OTH is a catchall of simple and complex lesions, it would be important to know what are the patient phenotypes that contribute to the significant risk findings – are these the patients with very complex lesions not easily classified into any one of the four lesion types used in this analysis or do they also include patients with isolated VSD/ASDs?

The list of Fyler terms and occurrence rates in each CHD phenotype category is now included in Supplemental Table 7. For example, the most common diagnoses in the OTH category include isolated secundum ASD (35%), partial anomalous pulmonary veins (12%) and sinus venosus atrial septal defect, superior type (10%).

-Please provide number of patients included in each of the risk analysis in Figure 1c, 2, 3. ie. please include n=XX for each outcome risk analysis shown in these forest plots. Please provide p values.

The number of patients in each of the risk analysis figures is now provided in Supplemental Table 4 and described in the Results, Figure legends, and Discussion.

We used a Bayesian approach to estimate relative risks. Unlike frequentist-based inferential statistical methods that rely on multiple test corrections and p-values, Bayesian statistical frameworks rely on probabilistic concepts that are especially well suited for estimating risk with

low sample sizes. In this context, Bayesian approaches provide a best estimate of relative risks and associations among genotypes, phenotypes, and outcomes given the data. The statistical support for risk estimates is quantified by resampling to provide a 95% confidence interval for each estimate.

To address the reviewer concerns, we added the following paragraph to the Discussion:

“Overall, the number of adverse events in probands with damaging genotypes in some clinical contexts was relatively low, despite an initial corpus of >2000 cases. It is well established that the Bayesian statistical framework is particularly well-suited for predictions when numbers are limiting¹; indeed, this was our motivation for employing this framework. Bayesian approaches allow for the incorporation of prior knowledge, explicitly model uncertainty, and provide established best-practice methodologies to avoid overfitting. Moreover, Bayesian methods model uncertainty using a probability distribution (the posterior distribution) for each parameter rather than a single point estimate. This last feature is particularly valuable in low-data scenarios, allowing the model to express uncertainty through conditional probability distributions even with limited data. This feature is particularly valuable in low-data scenarios, allowing the model to express uncertainty through conditional probability distributions even with limited data². The confidence intervals we report were generated by simultaneously resampling the data and rebuilding the net. This is a gold standard approach for estimating uncertainty when data are limiting. In general, the wide confidence intervals reflect this data scarcity. Taken together, these features allow Bayesian networks to make better informed predictions in low-data scenarios than do traditional frequentist approaches¹⁻³. It should be noted that risk projections for some of the rare, high-risk scenarios are rough estimates. While future analyses using larger cohorts will one day refine these estimates. Our results provide a necessary starting point for that future work.

The discussion for this data was focused on cardiac arrest for HTX patients, but I noted VENT outcomes for HTX patients yielded the highest risk, higher than for cardiac arrest. Given the finding of cilia and Foxj1 gene pathway enrichment for HTX patients, some discussion of the relevance of these findings to the increased adverse VENT outcomes should at least be mentioned.

As mentioned above, we added the following paragraph to the discussion:

“The ability to predict adverse outcomes allows for crucial, early deployment of potential therapeutic strategies to mitigate these risks and improve patient outcomes. For example, an extensive body of literature links ciliary dysfunction with respiratory complications, including prolonged ventilation, in the post-operative period^{6-9,16,17}. Pre-operative knowledge of a damaging cilia genotype would allow for early institution of aggressive airway clearance therapy, mucolytic therapies, inhaled β 2 agonists and specific ventilation strategies to promote mucociliary clearance¹⁶. Avoiding common anesthetic and analgesic agents that are known to

impair mucociliary function may also be beneficial in these patients¹⁷. As turn-around times shorten and costs decline for ultra-rapid and rapid WGS, genomic data will be available in the pre-operative period to allow for risk stratification and early interventions to mitigate adverse post-operative outcomes.”

In Figure 2, similar analysis was conducted for extracardiac anomalies. It would be useful to know what are the ECA recovered in the subjects with the pathogenic variants, and are there recurrent defects, such as limb anomalies, cleft palate, or renal defects. Could the authors comment on why there is such increase risk seen with ECA? Is this related to the ECA or simply that the complexity of birth defects reflects the central importance of the gene in multiple development pathways?

We agree with the reviewer that understanding the drivers of ECAs in increased adverse post-operative outcomes is an important question. Unfortunately, we don't have the power to address that question quantitatively, given the heterogeneity in ECAs and the low frequency of recurrent ECAs. Qualitatively and for clarity, we now include Supplemental Table 6 that describes the frequencies of ECAs in the cohort overall, in patients with damaging mutations in chromatin-modifying and cilia-related genes, and in patients that died.

-The presumed value of having genomic data and being able to assess risk for adverse outcomes following congenital cardiac surgery is to help inform postsurgical management of CHD patient care, thereby helping to improve postsurgical outcome. Please discuss how risk assessment based on having genomic data can improve postsurgical outcome for CHD patients. A brief consideration of how such knowledge might change clinical care needs to be addressed, as otherwise what is the point of knowing these risks?

We thank the Reviewer for this suggestion. We added the following paragraph to the Discussion:

“The ability to predict adverse outcomes allows for crucial, early deployment of potential therapeutic strategies to mitigate these risks and improve patient outcomes. For example, an extensive body of literature links ciliary dysfunction with respiratory complications, including prolonged ventilation, in the post-operative period^{6-9,16,17}. Pre-operative knowledge of a damaging cilia genotype would allow for early institution of aggressive airway clearance therapy, mucolytic therapies, inhaled β 2 agonists and specific ventilation strategies to promote mucociliary clearance¹⁶. Avoiding common anesthetic and analgesic agents that are known to impair mucociliary function may also be beneficial in these patients¹⁷. As turn-around times shorten and costs decline for ultra-rapid and rapid WGS, genomic data will be available in the pre-operative period to allow for risk stratification and early interventions to mitigate adverse post-operative outcomes.”

Reviewer #3 (Remarks to the Author):

In the manuscript ‘Genome Sequencing is Critical for Forecasting Outcomes following Congenital Cardiac Surgery’ the authors present genome sequencing data derived from patients with a congenital heart defect and show how genome sequencing has the potential to be used as risk stratification for children with congenital heart defects.

I would like to congratulate the authors for this very interesting manuscript which highlights the potential of genetic data in risk stratification of children with a congenital heart defect (CHD). Although this is an original and high-quality manuscript, I believe it could be improved in several ways.

1. Data may be presented in a more clear way:

Although the authors describe the CHD categories in an appropriate way, it is unclear why some patients have been classified in a certain category. For examples in supplemental table 8 in the AVC category, beside classic AVC pathology such as AVSD, there are patients with coarctation, aortic valve anomalies, transposition of the great arteries or truncus. Equally, some diagnosis seem to be classified into different categories (Ex: pulmonary valve stenosis or VSD appear in all categories). Could the authors explain.

Some cardiac defects are highly specific and occur at a high frequency in a single phenotype category while other defects occur at lower frequencies in multiple categories. We have added Supplement Table 7 that shows the frequency of Fyler terms by CHD phenotype. This pattern exists in the original physician-classified data obtained from the literature. We initially tried unsupervised clustering of patients using Fyler codes, but the clusters were diffuse and overlapping. We then turned to a gradient-boosted decision tree approach that learned the relative weighting of each feature for classification based on 3000 physician-classified patients. This method resulted in more accurate clustering that better reflected the physician-based phenotype assignments. We have added the following text to the Methods section to clarify this point.

“Considerable heterogeneity exists among Fyler descriptors and the five phenotype categories. For example, pulmonary stenosis and ventricular septal defects occur in all five categories. Other cardiac defects, such as heterotaxy, tetralogy of Fallot, hypoplastic left ventricle and others, are highly specific to a single phenotype category and occur at high frequency in that category. This data structure suggested that a supervised learning model that leverages prior physician classification information and weights each cardiac defect would perform better than an unsupervised classification approach. Therefore, we implemented a gradient-boosted decision tree

model trained on physician-based diagnoses to automatically classify each of the PCGC probands (14,765) into one of five CHD categories (see Supplemental Data, Supplemental Figures 1, 2).”

- *Supplemental tables 9 and 10 could benefit from having some clinical data added. I would suggest to add the underlying heart defect and category assigned.*

We have updated Supplemental Tables 1 and 2 (previously Supplemental Tables 9 and 10) to include the assigned phenotype category and the individual heart defects described by the Fyler terms.

- *In supplemental table 10, it is unclear whether the variants found, were found in homozygosity or heterozygosity for all patients. This is important to clearly mention, given that carrier state alone would probably not explain the CHD.*

We apologize for the ambiguity. We have updated Supplemental Table 2 (previously Supplemental Table 10) to indicate the bi-allelic state of every variant: either homozygous recessive or compound heterozygous, as damaging genotypes. Each row shows one variant and variants are sorted by the patient ID thus grouping compound heterozygous variants together in the table. Every patient in Supplemental Table 2 is either homozygous recessive or is a compound heterozygote.

- *I could not find the variables included for surgical outcome. In supplemental table 2, only mortality has been added. I might have missed it.*

We have clarified and simplified the variable names shown now in Supplemental Table 13 (previously Supplementary Table 2). The node names used in the Bayesian networks presented in the main text now match the variable headings in Supplemental Table 13. Additionally, we have now removed the surgical variables that were processed but not actually used in the final networks and risk predictions. Surgical variables either used or removed during the feature selection process are now listed in Supplemental Table 12.

2. Many variants have a VUS classification in ClinVar, how sure are the authors that these variants are disease causing? Has there any validation been performed to compare the results of GEM in relation to the ACMG classification?

Yes, the GEM method has been validated in a multicenter NICU study. We cite de la Vega et. al., (2021) which demonstrated that a GEM score cut-off of 1.0 detects 90% of true positives with a median of 2 reported candidate genes. We now include 4 additional publications using GEM to

report damaging genotypes. It is important to bear in mind that ACMG rules are used to classify a variant, whereas GEM identifies and prioritizes *genotypes*. These are different tasks.

3. In the same line, how sure are the authors that all found variants are causing the CHD phenotype. For example, a MYBPC3 variant in the context of CHD, could be an incidental finding depending on the underlying CHD.

We used a reporting criterion of a GEM score ≥ 1.0 . This threshold maximized our discovery rate while keeping our false positive rate low. Because GEM is phenotype aware, all reported genes should have at least some support in the literature for association with CHD. Damaged genes not related to CHD and associated with other phenotypes (e.g., PKU) were flagged as incidental and were not included in the paper.

To acknowledge that some genes are not necessarily key drivers of the CHD phenotype, we have included the following sentence in the Methods section:

“Because GEM is phenotype aware, reported genes have some support in the literature for association with the proband’s particular CHD phenotype. However, we cannot exclude the possibility that in some cases, the reported damaging genotype is incidental to CHD etiology; however, the high score threshold used for these analyses (> 1), means that there must have been strong phenotypic support for the genotype, and/or the variant(s) involved have strong ClinVar associations with pathogenicity”.

4. It’s remarkable that no associations have been found in patients with tetralogy of Fallot and AVSD. These categories of CHD have always been considered to have the largest genetic basis. Could the authors comment?

We agree with the reviewer’s comment and believe that these associations likely exist in this data set. For AVSD, we see a suggestive association with AVC and cilia genes, as reported in Table 1 and supported by existing literature. However, the small sample size leads to wide confidence intervals for this association, precluding statistical significance. We believe that an increase in sample size would tighten the confidence intervals and achieve statistical significance for the association of AVC with damaging cilia genes. We did not analyze CNVs/SVs genotypes in this analysis, given the nature of the whole-exome data. We suspect that inclusion of CNVs in this data set would show the well-established association between damaging CNVs and CTD, driven by TOF and chromosome 22q11 deletion.

5. Around 10% of CHD patients carried a (likely) pathogenic variant which could explain the phenotype. Although this is indeed in line with the published literature, it leaves the great majority (90%) of cases unexplained. It could be interesting that the authors add a word on future directions to solve these cases.

Thank you for this important point. We have added a sentence to the Discussion section addressing some of the new approaches that may improve our diagnostic rate, “New technologies, including long-read sequencing, reference-free whole genome assembly, and analysis of somatic tissues are likely to improve the diagnostic yield for CHD.”

References

1. Carlin, B.P. & Louis, T.A. *Bayesian Methods for Data Analysis, Third Edition*, (Chapman and Hall/CRC, 2008).
2. Kruschke, J.K. & Liddell, T.M. The Bayesian New Statistics: Hypothesis testing, estimation, meta-analysis, and power analysis from a Bayesian perspective. *Psychonomic Bulletin & Review* **25**, 178-206 (2018).
3. Gelman, A., *et al.* *Bayesian Data Analysis, 3rd edition*, (CRC Press, 2013).
4. Lee, S.Y. & Song, X.Y. Evaluation of the Bayesian and Maximum Likelihood Approaches in Analyzing Structural Equation Models with Small Sample Sizes. *Multivariate Behav Res* **39**, 653-686 (2004).
5. McNeish, D. On Using Bayesian Methods to Address Small Sample Problems. *Structural Equation Modeling-a Multidisciplinary Journal* **23**, 750-773 (2016).
6. Harden, B., *et al.* Increased postoperative respiratory complications in heterotaxy congenital heart disease patients with respiratory ciliary dysfunction. *J Thorac Cardiovasc Surg* **147**, 1291-1298 e1292 (2014).
7. Nakhleh, N., *et al.* High prevalence of respiratory ciliary dysfunction in congenital heart disease patients with heterotaxy. *Circulation* **125**, 2232-2242 (2012).
8. Stewart, E., *et al.* Airway ciliary dysfunction: Association with adverse postoperative outcomes in nonheterotaxy congenital heart disease patients. *J Thorac Cardiovasc Surg* **155**, 755-763 e757 (2018).
9. Swisher, M., *et al.* Increased postoperative and respiratory complications in patients with congenital heart disease associated with heterotaxy. *J Thorac Cardiovasc Surg* **141**, 637-644, 644 e631-633 (2011).
10. De La Vega, F.M., *et al.* Artificial intelligence enables comprehensive genome interpretation and nomination of candidate diagnoses for rare genetic diseases. *Genome Med* **13**, 153 (2021).
11. Kingsmore, S.F., *et al.* A genome sequencing system for universal newborn screening, diagnosis, and precision medicine for severe genetic diseases. *American journal of human genetics* **109**, 1605-1619 (2022).
12. Nyaga, D.M., *et al.* Benchmarking and quality control for nanopore sequencing and feasibility of rapid genomics in New Zealand: validation phase at a single quaternary hospital. *medRxiv*, 2024.2006.2013.24307636 (2024).
13. Ye, G.X., Ontiveros, E., Ivander, A., Velinov, M. & Simotas, C. Autosomal Recessive Infantile Hyaline Fibromatosis Identified Using Artificial Intelligence-Assisted Rapid Whole Genome Sequencing: A Rare, Multisystemic, Hereditary Disorder. *Cureus* **16**, e62037 (2024).
14. Miller, T.A., *et al.* Genetic and clinical variables act synergistically to impact neurodevelopmental outcomes in children with single ventricle heart disease. *Commun Med (Lond)* **3**, 127 (2023).
15. Jin, S.C., *et al.* Contribution of rare inherited and *de novo* variants in 2,871 congenital heart disease probands. *Nat Genet* **49**, 1593-1601 (2017).
16. Goetz, R.L., Vijaykumar, K. & Solomon, G.M. Mucus Clearance Strategies in Mechanically Ventilated Patients. *Front Physiol* **13**, 834716 (2022).
17. Feldman, K.S., *et al.* Differential effect of anesthetics on mucociliary clearance in vivo in mice. *Sci Rep* **11**, 4896 (2021).

18. Watkins, W.S., *et al.* De novo and recessive forms of congenital heart disease have distinct genetic and phenotypic landscapes. *Nat Commun* **10**, 4722 (2019).
19. Goldmuntz, E., *et al.* Frequency of 22q11 deletions in patients with conotruncal defects. *J Am Coll Cardiol* **32**, 492-498 (1998).

June 6, 2025

Author response to final reviewer queries.

Related to responses to Reviewer #2

1. Is the gene list used (partially or totally) extracted from the gene curation list of the CHD GCEP of ClinGen. Some of the authors are members of this group. If this is the case, maybe it should be stated somewhere in the text or referenced.

No, the gene lists used in the paper were curated by the authors directly from the literature or from the reactome.org database independently of CHD GCEP in ClinGen.

2. I find it hard to find table 2. I assume the authors mean table 1

Correct. We apologize for the typographical error and corrected the main text.

Reviewer #3

Although I believe that GEM as a system has been properly validated, I keep struggling with the large amount of VUS (based on ClinVar reporting) included. Looking at supplemental table 1, it seems that VUS get a lower GEM score than (likely) pathogenic variants. I would appreciate that the authors add a comment on this in their discussion.

We apologize for a labeling error in Supplemental Table 1 and thank the reviewer for catching this. The ClinVar column now correctly includes the terms ‘pathogenic’, ‘likely pathogenic’ and ‘not listed’, whereby variants that are not found in ClinVar are classified as ‘not listed’. We updated the variant discovery section in the Methods as follows:

“Notably, 48 of the 131 *de novo* variants identified by GEM were documented as pathogenic or likely pathogenic in ClinVar, and 27 of 48 (56%) of these were missense variants. The remaining 83 *de novo* variants were not listed in ClinVar, with 33 variants resulting in frameshifts, altered splice sites, or caused stop-gains, and 47 of 83 (57%) variants as missense variants. (Supplemental Table 1).”